# The role of cyclone activity in snow accumulation on Arctic sea ice

M.A. Webster [1]*, C. Parker [2,3], L. Boisvert[2] & R. Kwok[4]

Identifying the mechanisms controlling the timing and magnitude of snow accumulation on sea ice is crucial for understanding snow's net effect on the surface energy budget and sea-ice mass balance. Here, we analyze the role of cyclone activity on the seasonal buildup of snow on Arctic sea ice using model, satellite, and in situ data over 1979–2016. On average, 44% of the variability in monthly snow accumulation was controlled by cyclone snowfall and 29% by sea-ice freeze-up. However, there were strong spatio-temporal differences. Cyclone snowfall comprised ~50% of total snowfall in the Pacific compared to 83% in the Atlantic. While cyclones are stronger in the Atlantic, Pacific snow accumulation is more sensitive to cyclone strength. These findings highlight the heterogeneity in atmosphere-snow-ice inter-actions across the Arctic, and emphasize the need to scrutinize mechanisms governing cyclone activity to better understand their effects on the Arctic snow-ice system with anthropogenic warming.

[1] University of Alaska Fairbanks, Geophysical Institute, 2156 Koyukuk Drive, Fairbanks, AK 99775, USA. [2] NASA Goddard Space Flight Center, 8800 Greenbelt Rd., Greenbelt, MD 20771, USA. [3] Earth System Science Interdisciplinary Center, University of Maryland, 5825 University Research Court Suite 4001, College Park, MD 20740, USA. [4] Jet Propulsion Laboratory, 4800 Oak Grove Dr., Pasadena, CA 91109, USA. *email: mwebster3@alaska.edu

With the shift from thicker, older sea ice to younger, thinner ice in the Arctic[1–3], the timing and total accumulation of snow play increasingly important roles in sea-ice mass balance[4–6]. The high albedo of snow protects the sea-ice surface from solar radiation in spring and summer[7,8], while snow's insulating capacity hinders ice growth in autumn and winter, especially for thinner ice[4,9,10]. Thus, understanding the processes responsible for the seasonal build-up of the snow cover is fundamental for understanding snow's role in the Arctic sea-ice system[4]. Although snow accumulation on sea ice (e.g., ref. [11–15].) and Arctic cyclone activity (e.g., ref. [16–22].) have been investigated to date, previous works have not linked these phenomena together on multi-decadal and basin-wide scales currently possible with snow models, reanalysis data, and observations. Until now, the role of Arctic cyclone activity in snow accumulation on sea ice has remained an acute gap in our knowledge of snow in the sea-ice and climate systems.

In this work, we investigate the broad-scale relationship between cyclone activity and the seasonal build-up of snow on Arctic sea ice in 1979–2016 using reanalysis, in situ, and satellite data. A cyclone tracker is applied to reanalysis sea level pressure data to identify cyclones and examine the effects of their characteristics (e.g., count, intensity, size, and precipitation) on snow depth. We incorporate in situ information from ice mass balance buoys[23] to verify the primary mechanisms in snowpack formation. We survey the seasonal patterns of cyclone activity and snow depth over the entire Arctic basin and in nine distinct regions (Fig. 1a).

## Results

**Regional cyclone activity and snow depth.** Here, we present results for 1979–2016 from regional and seasonal perspectives. The seasons are defined as: autumn (September–November), winter (December–February), and spring (March–May). Building on the previous cyclone climatologies[18,19,24], our results show an emergence of two seasonalities split between the Atlantic and Pacific sides of the Arctic during the snow accumulation season

(September–May) (Figs. 2 and 3). Overall, cyclone activity is distinctly muted in the Pacific sector compared to the Atlantic sector (Fig. 2). Cyclone activity in the Pacific peaks in autumn and steadily declines in winter and spring, while in the Atlantic, cyclone counts, intensity, and snowfall reach their highest magnitudes in winter (Figs. 2 and 3).

The basin-scale pattern in cyclone-associated snowfall (Fig. 2g-i), together with regional differences in sea ice conditions, creates a similar spatial distribution in snow depth on sea ice, with more snowfall and deeper snow towards the Atlantic and less snowfall and thinner snow towards the Pacific (Fig. 4a-c). The Atlantic storm track[25] brings the heaviest snowfall (up to 100 + mm) during the snow accumulation season in the Barents, E. Greenland, and Kara seas (Figs. 2g-i, 3c, and 4d-f). Cyclone snowfall accounts for ~80% of the total snowfall in these regions (Figs. 2j-l and 3c), and coincides with a 25 + cm increase in snow depth between autumn and winter (Figs. 3e and 4d-f). In contrast, the snow cover in the Beaufort and E. Siberian seas undergoes modest increases of ~5–10 cm between autumn and winter (Figs. 3e and 4d-f) and cyclone snowfall accounts for ~50% of the total snowfall (Figs. 2j-l and 3d). These findings suggest that other mechanisms such as polar lows, trace precipitation, diamond dust, and rime ice, may play important roles in establishing the snow cover in the Pacific region. Ice advection may also influence snow conditions in the Beaufort Sea since multiyear sea ice is regularly advected from the Lincoln Sea[26], where snow accumulates earlier in the season[11,12].

Similar to airborne observations in the western Arctic[14,27], the reconstructed snow depths show deeper snow on multiyear sea ice than seasonal ice on average, but this relationship is more nuanced on regional scales. For example, the Chukchi and Kara seas both contain seasonal ice of the same average age[28], but the rate at which the snow cover builds in the Kara Sea is twice as high as that in the Chukchi Sea, resulting in a ~10 cm thicker snow cover (Figs. 3e and 4d-f). We attribute this difference to more frequent and intense cyclones associated with more snowfall in the Kara Sea compared to the Chukchi Sea (Figs. 2 and 3). The Atlantic Ocean is a large, open area of relatively warm water, serving as a vital moisture and energy source for cyclone

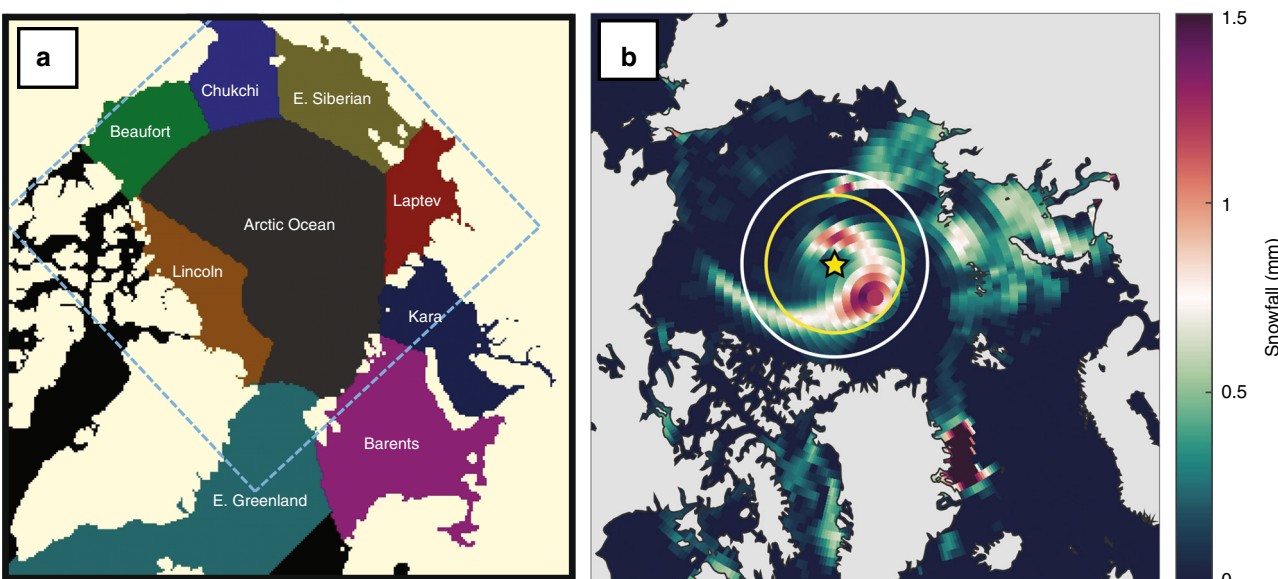

**Fig. 1** The regional breakdown of the Arctic domain and an example of a detected cyclone. **a** The segregation of regions defined in the analysis. The dashed square represents the domain of the snow depth reconstruction product (see Methods). **b** ERA-Interim snowfall fields, in mm, for 6 October 2013. An example of a detected cyclone is shown by the yellow circle. The central yellow star denotes the cyclone center. The yellow circle represents the mean radius used for cyclone size and depth. The white circle represents the 2-latitudinal degree additional buffer used to identify snowfall and precipitation associated with the cyclone event.

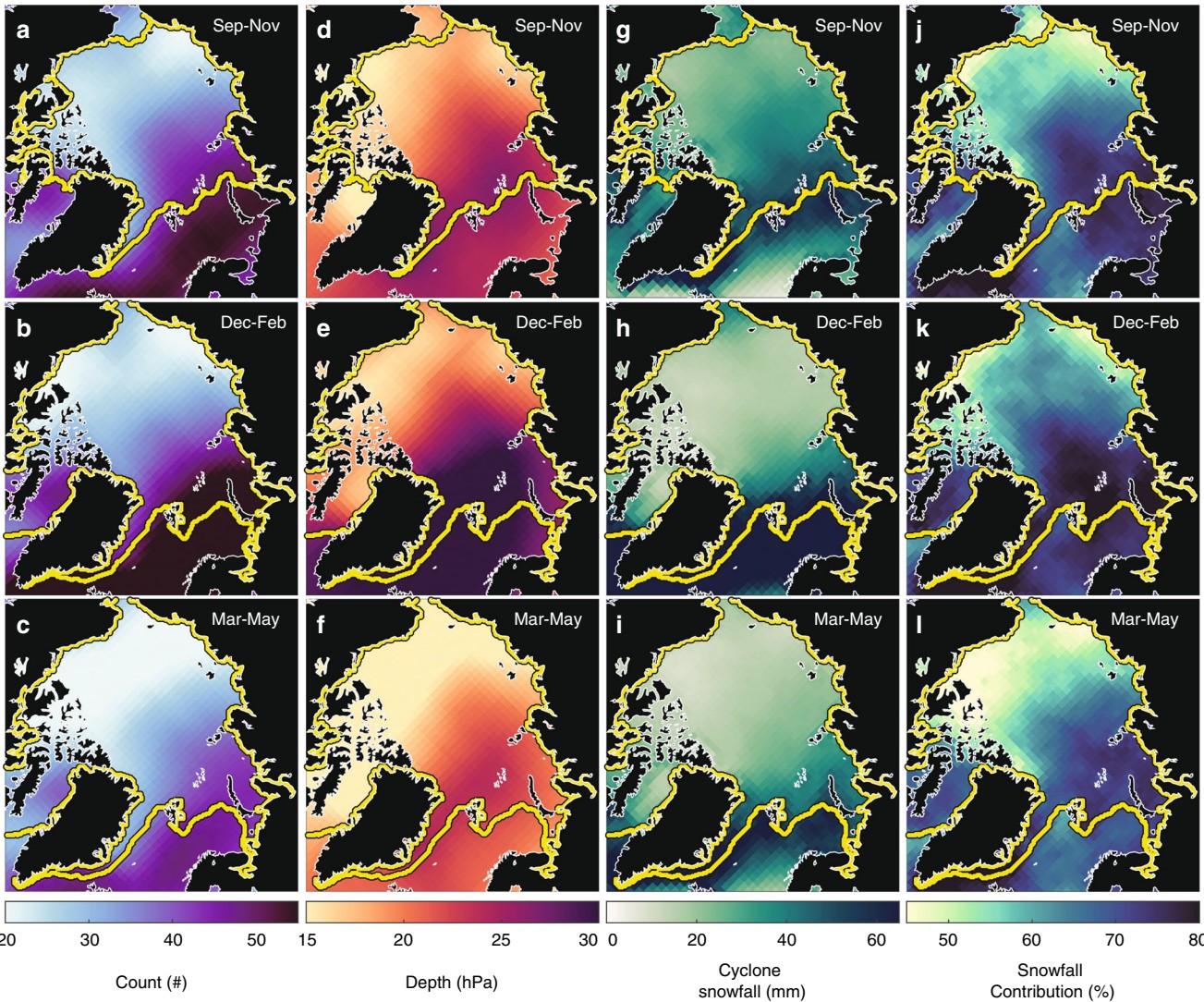

**Fig. 2** The 1979–2016 seasonal average of cyclone variables. Climatological average in: **a–c** cyclone event count, the number of times a grid cell is encompassed by a cyclone's area; **d–f** cyclone depth, a metric for cyclone intensity that takes into account cyclone size and intensity; **g–i** cyclone-snowfall amount; and **j–l** cyclone-snowfall contribution, the percent contribution of snowfall from cyclones relative to the total snowfall.

development along the Atlantic storm-track into the Arctic[25,29]. As shown in ref. [29], Atlantic storms advect more heat and moisture into the Arctic than Pacific storms.

Interestingly, the inter-annual variability between cyclone events, intensity, and snowfall does not spatially coincide (Supplementary Fig. 1). The Atlantic sector exhibited the greatest variability in cyclone counts (~10 events per season), but cyclone intensity was most variable in the E. Siberian and Lincoln seas. Cyclone snowfall was most variable along and southward of the ice edge in the Atlantic sector, whereas the cyclone-snowfall contribution was most variable over sea ice in the Pacific sector (Supplementary Fig. 1). The spatial discrepancies in the variability between cyclone variables indicate no clear relationship, likely culminating from a multitude of effects. For example, the inter-annual variability in other snowfall mechanisms, such as polar lows, frontal systems, and diamond dust, influences a cyclone's snowfall contribution since the latter is a function of total snowfall.

The inter-annual variability in snow accumulation (Fig. 4g-i) is also not spatially consistent with that in cyclone characteristics (Supplementary Fig. 1). The variability in snow accumulation may be partly influenced by the local sea ice conditions. As a case

example, there is a coincident localized area of high variability in snow accumulation (Fig. 4g-i) and date of closure (50-day standard deviation) in the western Laptev Sea (see Fig. 4 in ref. [28].). This shows that, on local scales, the timing of ice freeze-up may attenuate the effects of varying cyclone counts, intensity, and snowfall on snow accumulation.

Investigating the effects of freeze-up on snow accumulation further, we find that the average freeze-up date across all Arctic regions accounts for 29% of the variability in annual snow accumulation (Fig. 5a; Supplementary Table 1). However, this relationship varies from 1% in the Laptev Sea to 49% in the Barents Sea and Arctic Ocean. Compared to some peripheral seas (Kara, Chukchi, E. Greenland), the Lincoln Sea has earlier freeze-up and less seasonal ice, yet it has a stronger correlation between annual snow accumulation and freeze-up. The weaker relationship in some seasonal-ice regions may arise from several factors: first, total snowfall may be proportionally less in autumn than in winter and spring, and thus the annual snow accumulation may be less sensitive to the date of autumnal freeze-up; second, frequent cyclone events may occur throughout the accumulation season, overwhelming the effect of late freeze-up on snow accumulation; and third, there may be a later shift in the seasonal

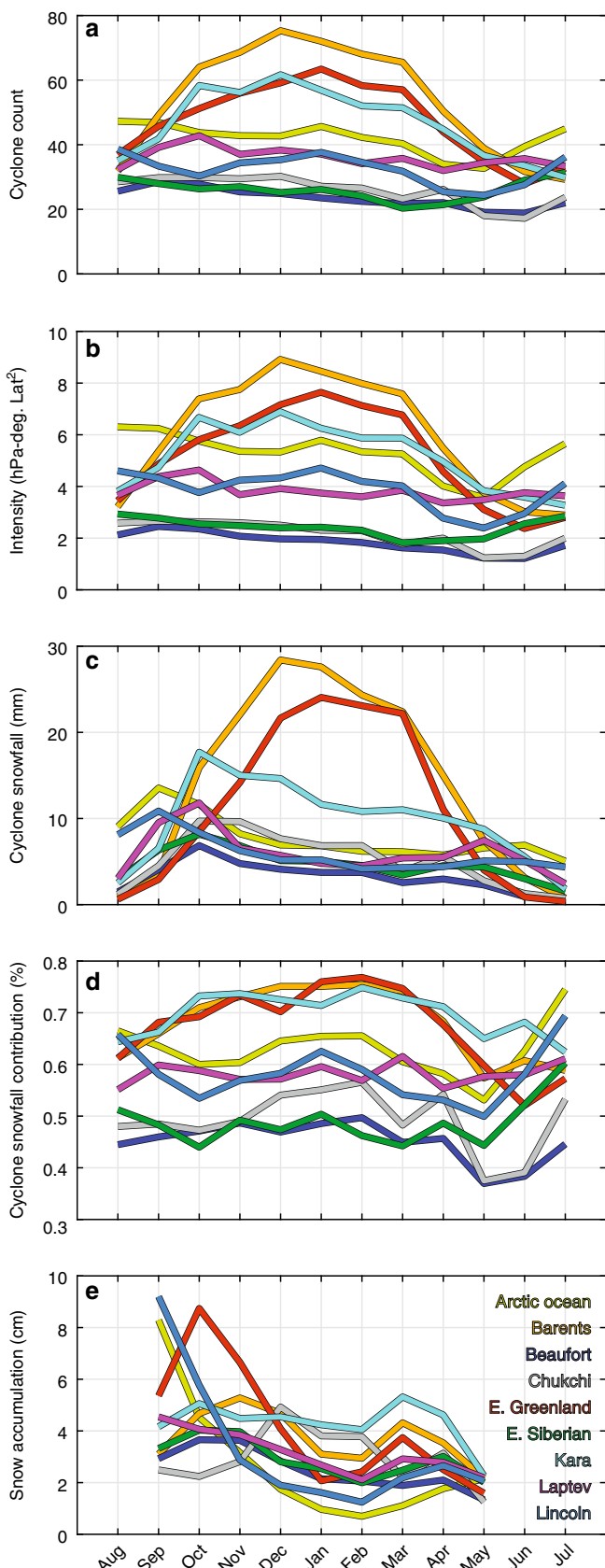

**Fig. 3** The 1979–2016 monthly mean in the seasonal cycle of cyclone variables and snow accumulation. Monthly averages of **a** cyclone count, **b** intensity, **c** cyclone snowfall, **d** cyclone-snowfall contribution, and **e** snow accumulation in nine distinct regions.

snowfall cycle[30], causing increased winter snowfall and subsequently compensate for the lack of snow accumulation in autumn.

**Build-up of the snowpack.** Using daily reconstructed snow depths, we identified the 1979–2016 average date at which 25%, 50%, and 75% of the snow cover was established (Fig. 6a–c). The date of maximum snow depth at a given location is used to back-calculate when 25%, 50%, and 75% of the maximum snow depth occurs. There is a strong latitudinal gradient in dates for 25% and 50% of the established snowpack (September–December), which is presumably driven by the waning duration of polar day and earlier sea ice freeze-up with increasing latitude. As the snow accumulation season progresses to 75% establishment, the range in dates across the Arctic shortens by half (Fig. 6a–c). For inter-annual variability, however, the range in dates lengthens as the snow season progresses, most notably in the seasonal-ice zone (Fig. 6d–f). The increasing spread is likely due to the combined effect of variable cyclone activity, sea-ice freeze-up[30], and the age and amount of snow-covered sea ice advected from other regions.

Figure 7 shows the cyclone count, intensity, and snowfall contribution of cyclone events leading to different proportions of the established snowpack. Cyclone count is the summation of the number of times a grid cell is encompassed by a cyclone area between September 1 and the first dates when 25%, 50%, and 75% of the maximum snow depth occur. Cyclone intensity and snowfall contribution are averaged from September 1 to the respective dates. For 1979–2016, there are typically fewer than 35 cyclone events (on average) when 25% of the snowpack is established. At 25%, the Arctic Ocean, E. Siberian, and western Lincoln seas experience the fewest number of events (~15 events), while the Barents and E. Greenland seas experience the most (~70 events). When 50% of the snowpack is established, there is a strong gradient in cyclone counts between the Pacific (~37 events) and Atlantic (~83 events) sectors, which is qualitatively similar to the distribution of cyclone counts in Fig. 2a–c. When 75% of the Arctic snowpack is established, the spatial gradient in cyclone events is at its most extreme, with a minimum of ~47 and maximum of ~120 events in the E. Siberian and Barents seas, respectively.

**Cyclone count vs. intensity.** Regarding the seasonal cycle in cyclone activity, our results raise the question: is the number of cyclone events or cyclone intensity more important for establishing the snowpack on Arctic sea ice? To address this question, we conducted univariate analyses between cyclone snowfall (dependent), intensity, and count (independent) and univariate and multivariate analyses between monthly snow accumulation (dependent) and cyclone variables (independent) over ice-covered areas (Fig. 5b). We refer readers to the Methods for more information.

Univariate analyses reveal that cyclone snowfall exhibits the strongest positive correlation with snow accumulation, and alone explains ~44% of its monthly variance (ranging 0–72% in the E. Greenland Sea and Arctic Ocean, respectively) (Fig. 5b and Supplementary Table 2). On average, cyclone count and intensity alone explain ~13% and ~19% of the variability in monthly snow accumulation, respectively, and regionally range by 0–35%. Cyclone size has the weakest relationship with monthly snow accumulation. Considering both cyclone count and intensity in a multivariate regression framework, these variables account for 7–35% of the monthly variability in snow accumulation, whereas, collectively, cyclone count, intensity, and size influence 9–35%. As demonstrated above, the effects of cyclone activity on snow

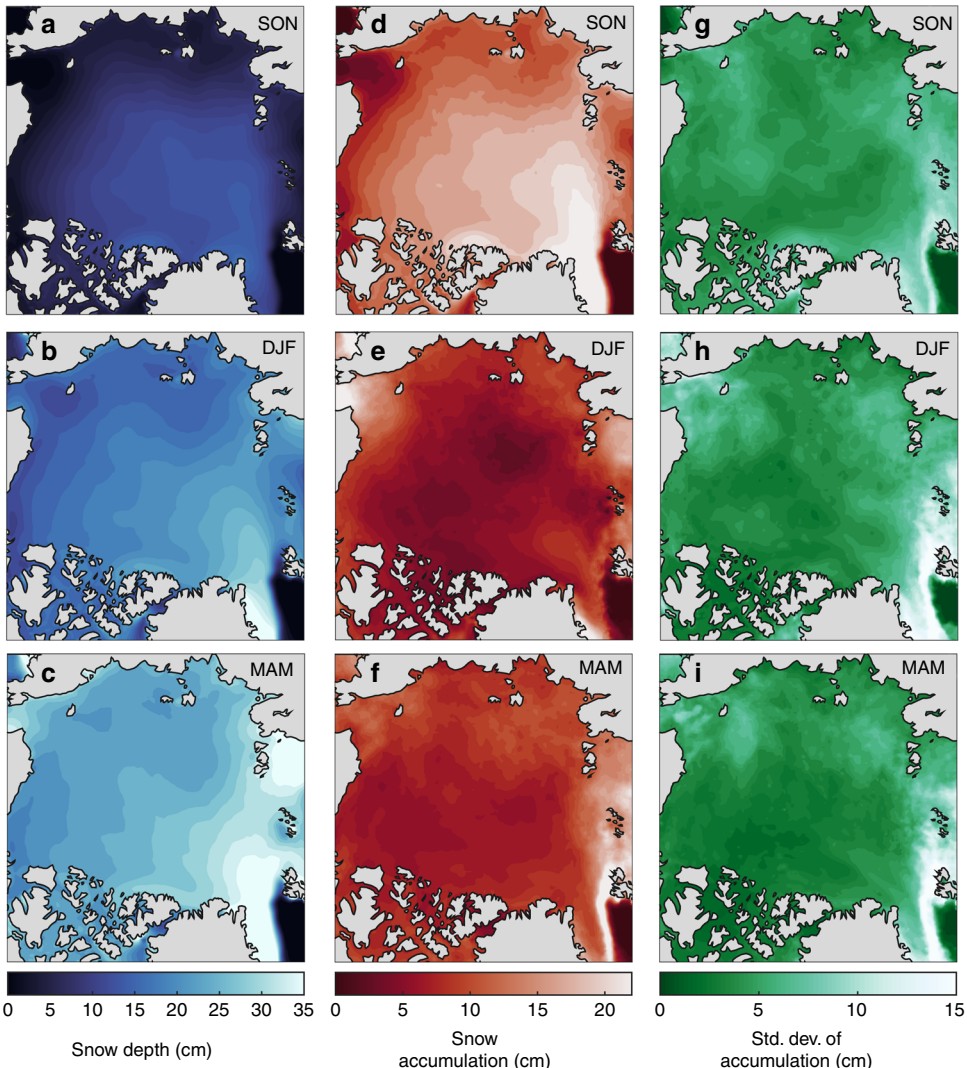

**Fig. 4** The 1979–2016 seasonal average of reconstructed snow conditions. Climatological values from the reconstructed snow depth product (see Methods) of: **a–c** snow depth; **d–f** snow accumulation (i.e., the difference in snow depth between the last and first day of the season); and **g–i** the standard deviation in snow accumulation. Note the differences in scales between snow depth, snow accumulation, and the standard deviation.

accumulation may be attenuated by the differing presence, advection, and date of freeze-up of sea ice between regions, and, thus, contribute to the extreme spread in the univariate and multivariate correlations.

Building on this, we compare the influence of cyclone count and intensity on snow accumulation between the Pacific and Atlantic sectors. The univariate analyses establish that, in the Pacific, cyclone intensity has ~10% greater influence on the monthly snow accumulation than cyclone counts (Fig. 5b and Supplementary Table 2). Similarly, cyclone-associated snowfall is more strongly correlated with intensity than cyclone count in the Pacific (Fig. 5 and Supplementary Table 3). The relatively higher correlation may arise because vorticity, which is proportional to the Laplacian of the sea level pressure fields used in this analysis for cyclone intensity, is a measure of strength of rotation and convergence/uplift. Convergence and uplift, in turn, transport moisture and heat from the surface into the Arctic atmosphere, where the moisture can then precipitate as snow. The statistics suggest that snowpack development in the Pacific sector is more sensitive to changes in cyclone intensity rather than cyclone counts. Increasing the frequency of weak cyclones may not significantly increase snowpack development in the Pacific sector; however, stronger cyclones could result in greater

snow accumulation. In the Atlantic sector, the results indicate that both cyclone intensity and count have roughly equal influence on monthly snow accumulation and cyclone-associated snowfall (Fig. 5; Supplementary Tables 2, 3).

**Other mechanisms of snow accumulation**. The relative importance of cyclone snowfall varies across the Arctic basin. Our analyses consistently reveal a thinner snowpack in the Pacific sector (particularly the E. Siberian and Beaufort seas, Fig. 4). On the first order, this can be explained by the low cyclone activity and weaker cyclones which account for ~50% of the total snowfall in the region (Figs. 2 and 3).

To explore the regional differences in snow accumulation and cyclone activity further, we analyzed snow depth data from ice mass balance buoys (IMBs)[23] in the broader Beaufort region (i.e., Beaufort Sea, western central Arctic) and the N. Atlantic region (i.e., E. Greenland Sea, eastern central Arctic), (Fig. 8). We used the IMB data together with cyclone variables to quantify how much of the snowpack is built-up during and outside of cyclone events. The Beaufort region had 19 buoys with 168 accumulation events, whereas the N. Atlantic region had 16 buoys with 246 accumulation events. Air pressure was recorded by the IMBs and

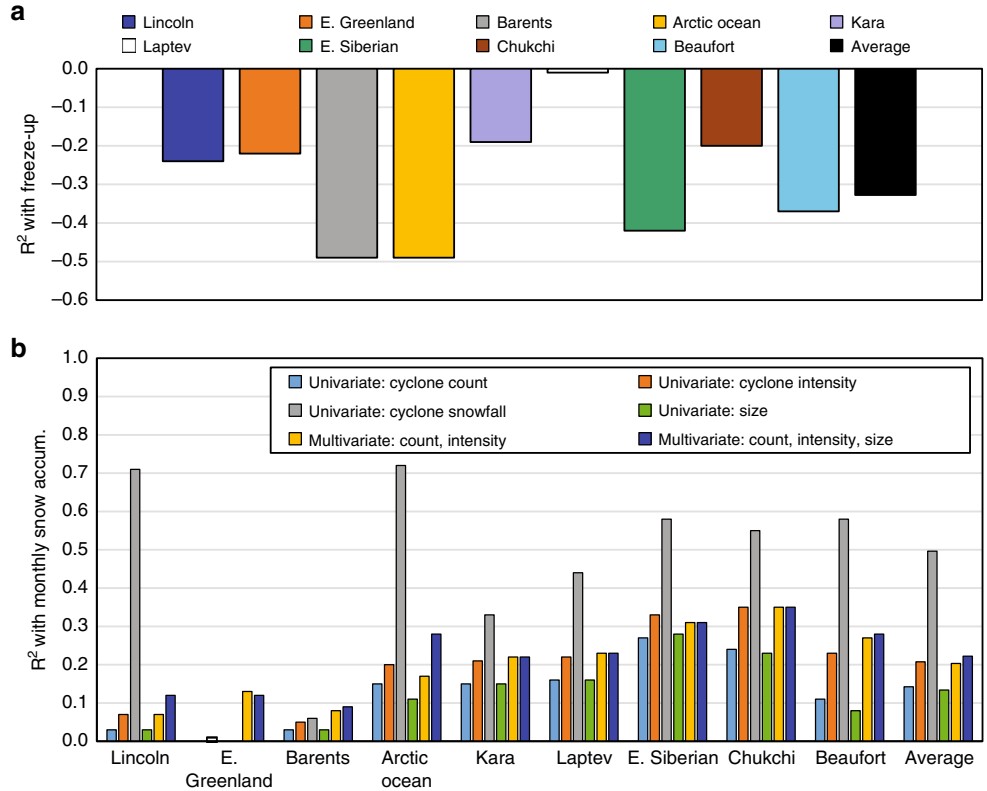

**Fig. 5** Correlations of snow accumulation with freeze-up dates and cyclone variables for 1979–2016. **a** The univariate linear regression between the timing of sea-ice freeze-up (independent) and the annual snow accumulation (dependent); **b** Univariate and multivariate correlations between monthly snow accumulation (dependent) and cyclone variables (independent) including: cyclone count, cyclone snowfall, and intensity during the snow accumulation season (September–May). In both panels, the solid colors represent at least 95% statistical significance, while hatched colors are not statistically significant. Cyclone variables are only considered in areas with a 15% sea-ice concentration or greater.

compared to cyclones' minimum sea level pressures. A 24-hour window and minimum cyclone radius were used to identify coincident events between the IMB buoys and cyclone tracker.

According to the IMB data, just under half of the snow accumulation events in the broader Beaufort region were associated with cyclones, while ~64% of the accumulation events in the N. Atlantic region corresponded with cyclones. Both quantitatively and qualitatively (Fig. 8), the snow accumulation events in the Beaufort region occur more frequently from other mechanisms that are not captured by the reanalysis product and cyclone tracker used in this study[31,32]. In general, there was better agreement in the air pressure between the IMBs and cyclone events in the N. Atlantic than in the Beaufort region. Accumulation events in the N. Atlantic coincided with more intense, well-developed cyclones, as evidenced by the IMBs' air pressure readings. Although cyclone trackers and reanalysis data have inherent uncertainties, these findings underscore that snowfall from other mechanisms may be equally important as cyclone-snowfall in establishing the snowpack in the Beaufort region (Figs. 7 and 8). Such snowfall may be associated with polar lows, rime ice, trace precipitation, or come in the form of diamond dust, the latter of which previous field observations have attributed ~2–3 cm of annual snow accumulation to[11].

**Inter-decadal changes and variability**. We investigated the trends in snow depth and cyclone characteristics over 1979–2016 to determine possible linkages between their long-term changes. For a full review of interpreting trends in cyclones using different reanalysis products and trackers, we refer readers to ref. [33,34]. We

evaluated the average of March and April snow depths and the regional sums and averages of cyclone variables and precipitation over the snow accumulation season (September–May), independent of sea ice area.

Positive trends in snow depths occurred in the Lincoln and E. Greenland seas, with the latter having a statistically significant increase of 2.4 cm per decade (Fig. 9). Although all statistically insignificant, the increase in snow depth in the Lincoln Sea was accompanied by an increase in total snowfall, cyclone snowfall, cyclone counts, and intensity (Table 1). In the E. Greenland Sea, there were no statistically significant changes in cyclone variables that readily explain the positive trend in snow depth. Total snowfall and cyclone snowfall decreased while total rainfall and cyclone rainfall increased over 1979–2016 (Table 1).

In contrast to the Lincoln and E. Greenland Seas, all other regions exhibited a decline in snow depth over 1979–2016 (Fig. 9). The region with the largest negative trend was the Barents Sea with −3.8 cm per decade, followed by the Kara and Chukchi seas with −2.0 and −1.7 cm per decade, respectively. While different factors contribute to snow loss in different regions, there is an emerging signal of statistically significant changes in precipitation phase and freeze-up date that may influence the decline in snow depth. In the Barents Sea, less total snowfall occurred over 1979–2016, and a greater proportion of cyclone precipitation came in the form of rainfall (Table 1). The Barents Sea also experienced a trend of later first (early) freeze-up by 13 days per decade. Without sea ice present as a platform for snowfall to accumulate on, snow falls directly into the ocean[3].

In the Chukchi Sea, the trends in cyclone variables were not statistically significant and do not correspond with the decline in

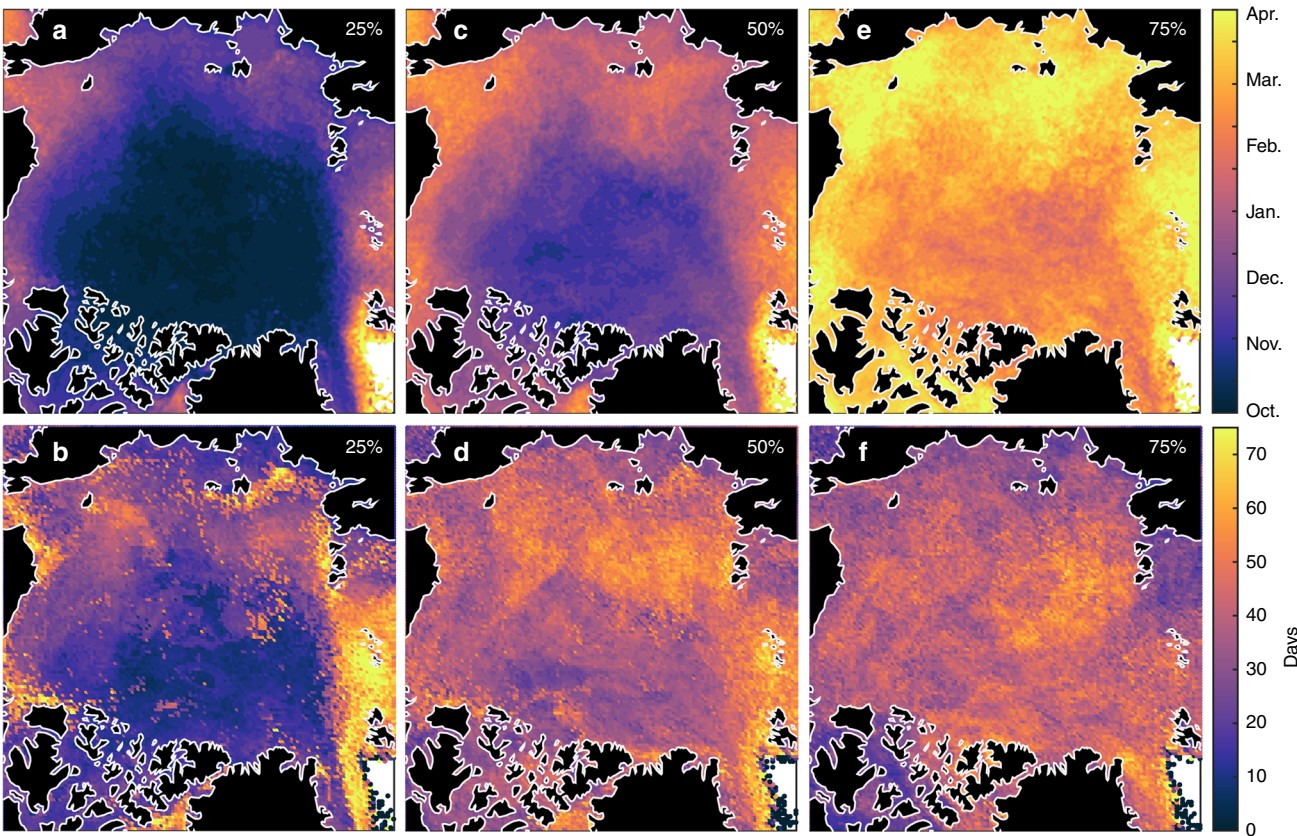

**Fig. 6** Dates of when different proportions the snowpack are established. The 1979–2016 average dates on which: **a** 25%, **c** 50%, and **e** 75% of the snow cover is established. The standard deviation (days) of the 1979–2016 dates for the corresponding **b** 25%, **d** 50%, and **f** 75% snow depth thresholds. The standard deviation indicates which regions are susceptible to low (Central Arctic) and high (East Greenland and Barents seas) variability in snow depth conditions due to cyclone activity, the timing of sea ice freeze-up, and sea ice advection.

snow depth. However, the observed delay in sea-ice formation was significant: early (first) freeze-up occurred 15 days per decade later in 1979–2016, whereas late (continuous) freeze-up occurred 13 days per decade later (Table 1). We note that cyclone snowfall only accounts for ~50% of total snowfall in the Pacific region and thus changes in other precipitation mechanisms may contribute to the negative trend in snow depth in the Chukchi Sea, as well as in other regional seas.

## Discussion

In this study, we analyzed the relationships between the seasonal build-up of snow on sea ice and cyclone events, intensity, cyclone-related snowfall, and sea-ice freeze-up. The spatial and temporal variations in cyclone activity and their contributions to snow on Arctic sea ice have not been previously characterized seasonally or regionally with respect to the snow accumulation season. The heterogeneous nature of our findings, both spatially and temporally, highlights the complexity of the snowpack response to cyclone activity and local sea-ice conditions.

A key result from our work is that cyclones account for ~80% of total snowfall in the Atlantic regions. These regions appear to be more reliant on cyclone snowfall to establish a snowpack. Conversely, cyclones account for only ~50% of total snowfall in the Pacific region and therefore snow input from other mechanisms may be equally important for snowpack establishment. There are factors to consider when interpreting this result, specifically regarding missed snowfall during cyclone events and systematic trace precipitation. Not all cyclone snowfall may be captured by the spatial buffer in the tracking methods, in

particular, the entire comma-like pattern of cyclone precipitation (Fig. 1b). Equally important, trace precipitation, or residual drizzle, is a known issue in reanalysis products, which results in too frequent and too much snowfall outside of true snowfall events (Supplementary Figs. 2 and 3)[35,36]. Thus, both the missed cyclone snowfall and trace precipitation may erroneously increase the proportion of snowfall outside of cyclone events and lessen the amount that cyclones truly contribute. The effects of such biases necessitate the collection of process-oriented observations of cloud microphysics and precipitation to improve understanding of precipitation mechanisms and their treatment in models.

To assess ERA-Interim snowfall and precipitation, we compared these variables with those from NASA's Atmospheric Infrared Sounder (AIRS)[37] and CloudSat[38] (Supplementary Figs. 2 and 3). Throughout the accumulation season (September–May), ERA-Interim and CloudSat produce similar snowfall, with ERA-Interim producing an average of $1.20 \pm 0.08$ mm-day$^{-1}$ over the sea-ice pack (defined as 15% ice concentration and greater) compared to $1.11 \pm 0.10$ mm-day$^{-1}$ from CloudSat. When total precipitation from ERA-Interim and AIRS were compared, AIRS produced less precipitation over sea ice, with ERA-Interim having an average of $1.93 \pm 0.13$ mm-day$^{-1}$ compared to $1.31 \pm 0.10$ mm-day$^{-1}$ from AIRS. In both comparisons, ERA-Interim produces more snowfall and total precipitation, which underscores the possibility that trace precipitation is over-produced in the reanalysis product and leads to an underestimation of cyclone contribution to snow accumulation on sea ice.

The second and third key findings from this analysis are that the role of cyclone activity in snow accumulation varies greatly

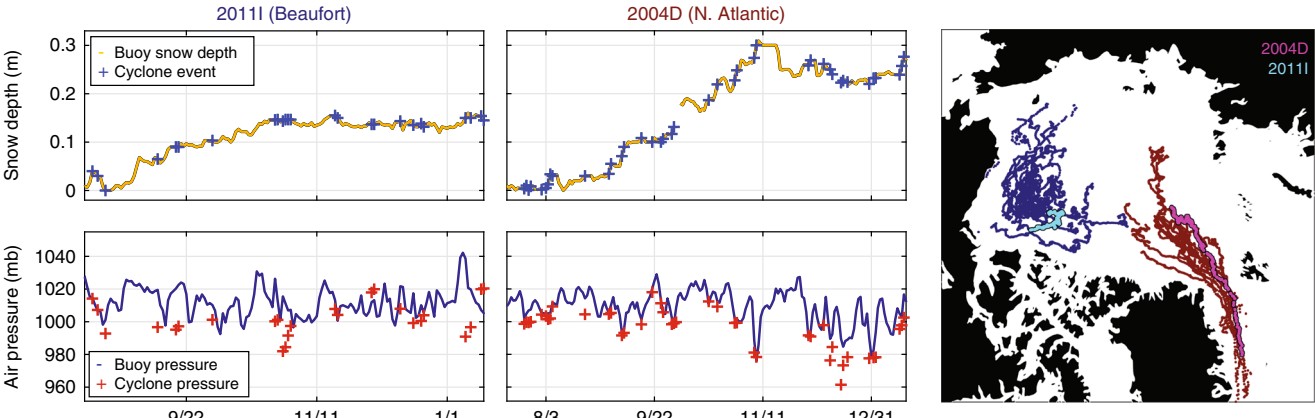

**Fig. 7** Cyclone characteristics as different proportions of the snowpack are established. The cyclone event count (i.e., the number of times a grid cell is encompassed by a cyclone's area), cyclone depth (a metric for intensity that takes into account both cyclone size and intensity), and cyclone-snowfall contribution for **a–c** 25%, **d–e** 50%, and **g–i** 75% of the established snow cover with respect to the 1979–2016 mean snow depth.

**Fig. 8** Cyclone events and snow depth detected by the tracker and buoys. In the Beaufort region, 48% of the snow accumulation events from 19 buoys were associated with cyclones. A typical example of accumulation events is shown by buoy 2011I, and its drift track is in cyan on the map. In the North Atlantic, 64% snow accumulation events from 16 buoys were associated with cyclone events. A typical example of accumulation events is shown by buoy 2004D, with its drift track shown in pink on the map. Note the difference in snow depth changes and air pressure between buoys, with 2011I showing smaller snow depth changes and higher air pressure readings, which is indicative of weaker systems, than those recorded by 2004D.

and local sea-ice conditions can attenuate the effects of cyclones. In the Pacific sector, cyclone snowfall and snow accumulation were more sensitive to cyclone intensity than cyclone counts, while both intensity and counts had equal influence in cyclone snowfall and snow accumulation in the Atlantic (Fig. 5b and Supplementary Tables 2, 3). Sea-ice freeze-up dates alone account for 29% of the variability in annual snow accumulation, and regionally range from 1% (Laptev Sea) to 49% (Barents Sea and Arctic Ocean). The delay in sea-ice freeze-up corresponds to the decreasing trends in snow depth in most regions (Fig. 9, Table 1). Compared to observations[18], the negative trends in snow depth determined here are smaller and have a weaker relationship with sea-ice freeze-up. One factor contributing to this discrepancy is the survival and accumulation of snow during summer, which occurred in earlier decades but, for contemporary times, melts away in summer (see Supplementary Fig. 4).

To conclude, our study brings new insight on the role of cyclone activity in snow accumulation on Arctic sea ice. There are several directions in which future analyses can build on this work. The survival and accumulation of snow during the melt season has not been quantified (or linked to cyclone activity), yet this has critical consequences on the radiative energy balance, long-term changes in snow depth (i.e., Fig. 9), and the retrieval of sea ice thickness from altimetry data (e.g., ref. [39]). The shift to an earlier melt season[28,40] also has important implications for snow on sea ice and the snow-albedo feedback (e.g., ref. [41]), yet this relationship has not been fully investigated via field and remotely-sensed observations. Equally, the response of cyclone activity to a warming climate is highly relevant to understanding long-term changes in snow conditions on sea ice, especially with

regard to more frequent rainfall[42], and remains a topic of further investigation.

## Methods

**Cyclone tracker.** The Melbourne University cyclone tracking scheme was selected for the analysis[20,43–45] due to its consistency in capturing cyclone events[20,33,46,47], its broad agreement in results with other cyclone tracking algorithms[33,34], and its ability to detect open and closed systems (both of which produce snowfall over sea ice). Six-hourly data of total precipitation, snowfall, and sea level pressure (SLP) from ERA-Interim[35] were selected for this analysis due to the good agreement between ERA-Interim and AIRS and CloudSat observations (Supplementary Fig. 3), in situ data from Arctic coastal stations, drifting ice stations and ice mass balance buoys[36,48,49], as well as for its availability for the 1979–2016 period. The cyclone results derived from ERA-Interim with the Melbourne University cyclone tracker and other cyclone trackers are well within the spread in cyclone parameters (e.g., count, track, intensity, trend) derived from other reanalysis products, as shown in ref. [33,34,50].

Our study applies the Melbourne University cyclone tracking scheme, a quasi-Lagrangian tracker, in a Eulerian framework[40,51], which provides greater insight into the effects of cyclone events on the snowpack. For example, if a cyclone tracks over the same location twice, our method counts this as two events, since the snow cover could experience two events of cyclone-related snowfall. More traditional Lagrangian approaches would account for one event in these instances.

Before running the cyclone tracker, we converted the ERA-Interim sea level pressure, total precipitation, and snowfall data in geodetic coordinates to Polar Stereographic coordinates, which were then interpolated using a bicubic spline to create a one-latitudinal degree grid centered on the North Pole[52,53]. A weighted spatial filter was applied to average the SLP over two latitudinal degrees to eliminate small-scale features susceptible to erroneous cyclone identification[53]. The Laplacian of the SLP fields were then calculated to determine the local maxima of Laplacian relative to eight neighboring grid cells. Once these local maxima are identified, a set of criteria was imposed: first, the second derivative of the SLP in the x- and y-directions must be positive;[52] and second, the mean Laplacian in the immediate vicinity of the maxima must meet the concavity criterion where the Laplacian is equal to or greater than 0.2 hPa per degree latitude squared[20]. At every 6-hourly time-step, the cyclone centers were determined through an iterative approach that finds the minimum first derivatives (in x and y) within the local area of a center candidate, identifying both open and closed systems[52]. With the exception of the analysis involving IMB buoys, the radii of cyclones were taken as the mean radial distance in which the Laplacian reaches near-zero. We apply an additional 2-latitudinal degree buffer to the radial distance to encompass a greater proportion of the precipitation associated with cyclones, especially for those exhibiting an asymmetric, comma-like shape (Fig. 1b). Given that IMB buoys are point measurements, we apply the minimum radial distance in the tracker and use a 24-hour window to more accurately identify and isolate coincident events between the cyclone tracker and IMB buoy data.

Following ref. [20], cyclone intensity was measured through SLP, the Laplacian of the SLP fields (which is proportional to vorticity), and depth which is a function of the squared radius and mean Laplacian in the immediate vicinity of the cyclone center. For all results, intensity refers to the Laplacian of the SLP fields in hPa-deg lat² unless otherwise stated. All points within the radius of a cyclone were identified as part of the same cyclone event. Cyclone events were counted at each six-hourly time-step and divided by four to yield the count per day over each grid cell. For more details on the Melbourne University cyclone tracker, we refer readers to ref. [20,52]. Note, our values in cyclone characteristics are comparatively larger than those in ref. [20]. due to the Eulerian framework, mean radius, and the application of a 2-latitudinal degree buffer in the tracker for precipitation.

**Ice mass balance buoys.** Daily averages of snow depth data from 35 ice mass balance buoys (IMBs)[23,54] for 2000–2015 were evaluated for snow accumulation events. These data were used to determine the number of accumulation events coinciding with cyclones detected by the cyclone tracker. These IMBs are listed in

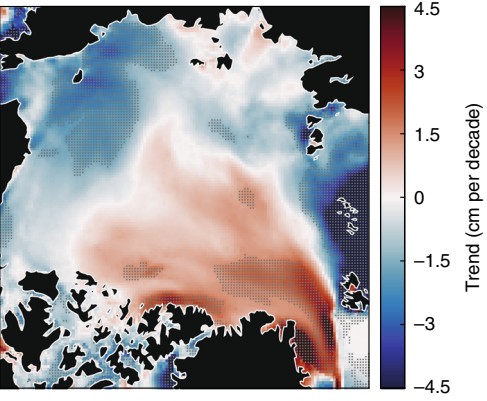

**Fig. 9** The trend in snow depth (centimeters per decade) for the 1980–2016 March/April average. March and April were selected since they represent the annual snow depth maximum for the majority of Arctic sea ice. The dots indicate areas with at least 95% statistical significance.

**Table 1 The 1979–2016 regional trends in snow depth, cyclone variables, precipitation, and sea-ice freeze-up per decade.**

|  | Snow depth (cm) | Cyclone count (#) | Cyclone intensity (hPa-deg Lat.²) | Cyclone rainfall (mm) | Cyclone snowfall (mm) | Cyclone precip. (mm) | Total rainfall (mm) | Total snowfall (mm) | Total precip. (mm) | Early freeze-up (days) | Late freeze-up (days) |
|---|---|---|---|---|---|---|---|---|---|---|---|
| Arctic O. | −0.1 | 4.9 | 0.081 | **0.001*** | **0.001*** | **0.002*** | **0.001*** | **0.001*** | **0.002ᵃ** | 5* | 4* |
| Barents | **−3.8*** | 1.8 | −0.012 | **0.008*** | **−0.006*** | 0.002 | **0.008*** | **−0.005*** | 0.003 | 13* | 7* |
| Beaufort | **−0.8*** | 2.1 | 0.020 | 0.003 | 0.000 | 0.003 | **0.004*** | 0.000 | 0.004 | 10* | 9* |
| Chukchi | **−1.7*** | 0.6 | 0.005 | 0.004 | 0.001 | 0.005 | **0.007*** | 0.003 | **0.010*** | 15* | 13* |
| E. Greenland | **2.4*** | 1.1 | −0.021 | **0.005*** | −0.005 | 0.000 | **0.005*** | **−0.005*** | 0.000 | −1 | 1 |
| E. Siberian | **−1.5*** | 4.7 | 0.022 | 0.002 | **−0.001ᵃ** | 0.001 | 0.003 | −0.001 | 0.003 | 13* | 11* |
| Kara | **−2.0*** | −6.9 | −0.106 | 0.003 | 0.001 | 0.004 | 0.004 | 0.003 | 0.008 | 13* | 9* |
| Laptev | **−0.7*** | −3.6 | −0.038 | 0.003 | 0.004 | 0.007 | 0.004 | 0.006 | 0.010 | 9* | 8* |
| Lincoln | 0.5 | 6.1 | 0.111 | **0.001*** | 0.005 | 0.006 | **0.002*** | 0.005 | **0.007*** | 0 | 1 |

Bold text and asterisks indicate at least 95% statistical significance. Rainfall, snowfall, and precipitation amounts were normalized by the corresponding regional areas to allow inter-comparison across regions. Note, cyclone and precipitation variables were not normalized by the sea-ice area due to the negative trends in sea-ice extent.
ᵃVariables subject to autocorrelation

Supplementary Table 4. IMBs are equipped with a sonic range-finder which detects the snow (or ice) surface at four-hourly intervals. Given instrumental uncertainty and precision, we defined accumulation events as an increase in snow depth greater than one centimeter within a 24-hour window[23]. The data were quality checked for erroneous values due to changes in buoy orientation, the formation or removal of large snow drifts, and false GPS readings. While this check improves the data quality, some detected changes in surface elevations may be due to wind-driven redistribution of snow. The reader is referred to ref. [55]. for additional details on the quality check.

**Sea-ice freeze-up**. Sea ice freeze-up dates derived from passive microwave data[40] were used to evaluate factors that influence the inter-annual variability in snow depth on sea ice. We selected continuous freeze-up dates rather than intermittent freeze-up dates due to the effects of melt on snow depth. Dates for autumn 1979–2015 correspond with snow depths from autumn 1979–2015 to the following spring 1980–2016 in the analysis since the timing of autumnal freeze-up can have long-lasting effects on snow accumulation.

**Snow depth reconstructions**. Daily snow depths were reconstructed following ref. [56]. The approach synthesizes ERA-Interim reanalysis snowfall, sea ice motion[57], and Bootstrap sea ice concentration[58] data to estimate the accumulation of snow on drifting ice parcels when sea ice concentrations are 50% or higher and 2-m air temperatures are below freezing. To convert snowfall to snow depth, scaled climatological densities from ref. [12]. were applied[56]. Initial snow depth values were set to zero on September 1 of each year to isolate the effects of cyclone activity on snow accumulation (Supplementary Fig. 4). The spatial domain of the reconstruction is smaller than the cyclone results, and is outlined by the dashed blue line in Fig. 1a. In this analysis, we define snow depth as the reconstructed value and snow accumulation as the difference in snow depth between the first and last time-step over a given duration of interest (e.g., monthly, annual). However, for the spring season, accumulation is calculated as the difference between the maximum snow depth and March 1 due to sea-ice retreat in May (which would otherwise yield negative snow accumulation). Snow depth for a given season is averaged.

Snow depth reconstructions do not account for wind-driven snow redistribution, ice deformation, blowing snow lost to leads, or melt. The combination of these processes may contribute to potential biases and uncertainties in the snow depth reconstruction on a seasonal basis. However, the absence of loss terms enables the direct assessment of the relationship between cyclone activity and snow accumulation on sea ice. Comparisons have been made with snow depth retrievals from Operation IceBridge's snow radar[59] and ice mass balance buoys[55]. Both analyses found generally good agreement between the snow depth reconstruction and airborne snow depth retrievals. On a broad-scale view (Fig. 4a-c), the 1979–2016 snow depth reconstruction is in good agreement with the 1954–1991 snow depth climatology[11,12,60], which suggests that the large-scale distribution of reconstructed snow depths may be suitably representative.

**Statistical approach**. To evaluate the relationship between cyclones and snow accumulation on sea ice, we used univariate linear regressions between the monthly snow accumulation (dependent) and monthly cyclone characteristics (independent) over sea ice. To validate the choice of a linear regression model, we examined linear[61,62], cubic, and exponential regressions for all variables across all regions. Linear regressions yielded the lowest root-mean-squared (RMS) errors for the majority of cases (see Supplementary Information), demonstrating the strongest fit among variables. For consistency across variables and regions, we therefore deferred to using a linear regression model.

The analysis includes all variables in September–May for 1979–2016, has a sample size of 336, and uses 0.05 for 95% statistical significance (0.05 is used across all statistical analyses). A monthly temporal resolution was chosen to avoid the high-frequency fluctuations in daily snow depth due to ice motion. The cyclone characteristics are calculated as the monthly regional mean of the sum of: count; intensity; size; and snowfall over sea ice. The monthly sum is taken from the last day of the preceding month to the penultimate day of the relevant month. This one-day offset accounts for the lag of snow accumulation on sea ice following cyclone events.

Using monthly variables over sea ice, we further evaluated the combined effect of cyclone characteristics on snow in a multivariate framework using a sample size of 336. The multivariate models use monthly snow accumulation (dependent) and cyclone count and intensity; and count, intensity, and size (independent). The models use the regional mean of the temporal sum of cyclone count, but the regional mean of the temporal mean in cyclone intensity and size to avoid double counting cyclones and erroneously estimating their effect on snow accumulation. Scatterplots in Supplementary Fig. 5c,g, h demonstrate that the cyclone variables considered in the multiple regression framework largely do not exhibit collinearity. This suggests that more cyclone events do not equate to stronger or larger cyclones and establishes that these variables can be treated as independents. Cyclone snowfall was not included in the multivariate analysis to avoid multicollinearity (Supplementary Fig. 5a, e). The multivariate models are assessed using the adjusted squared correlation coefficients to account for more degrees of freedom.

Given the strong positive correlation between cyclone snowfall and snow accumulation (see Fig. 5b and Supplementary Table 2), we analyzed the relationship between cyclone count and intensity and cyclone-associated snowfall. Regional averages of the monthly sums in these variables were used, resulting in a sample size of 456. Supplementary Fig. 5a, e demonstrates the positive correlations between cyclone snowfall, count, and intensity and a strong fit of the linear regression model.

Prior to the 1979–2016 trend analysis, we tested for autocorrelation among variables using the Durbin-Watson Test. The test reveals that, despite a few exceptions, cyclone counts, cyclone snowfall, intensity, total snowfall, and total precipitation are not autocorrelated. The exceptions are: cyclone snowfall in the E. Siberian Sea and total precipitation in the Arctic Ocean (autocorrelated); total snowfall in the Arctic Ocean and Laptev Sea and intensity in the Arctic Ocean (inconclusive). Furthermore, we use zero initial values in the snow depth reconstruction each year, which removes any carryover of snow from the preceding year. Therefore, with a few exceptions, the assumption of a zero serial correlation is justifiable for variables in the 37-year time-series. We examined linear, cubic, and exponential regressions for all variables across regions and chose linear regressions due to the strongest fits for exploring whether discernable trends in variables exist. We note that, for several variables, the trends were statistically inconclusive. We found no significant or coherent relationships between freeze-up and cyclone characteristics. Collinearity between these variables is unlikely to affect the interpretation of the linear regressions and inter-decadal trends.

## Data availability

ERA-Interim[44] data are available on the European Centre for Medium-Range Weather Forecasts website: https://www.ecmwf.int/. Sea ice motion for the ice parcels was derived from passive microwave brightness temperatures from the Scanning Multichannel Microwave Radiometer 37 GHz channel, Special Sensor Microwave/Imager 85.5 GHz channel, the Special Sensor Microwave Imager/Sounder 91.7 GHz channel, and Advanced Microwave Scanning Radiometer-EOS 89 GHz channel[57] and are available at http://rkwok.jpl.nasa.gov. The Bootstrap sea ice concentration data are available at https://nsidc.org/data/nsidc-0079. The raw ice mass balance buoy data[54] are available on the Cold Regions Research and Engineering Laboratory-Dartmouth website: http://imb-crrel-dartmouth.org/. The remaining datasets generated and/or analyzed during the study are available from the corresponding author on reasonable request.

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

## Acknowledgements

M.W., C.P., and L.B. were funded by the National Aeronautics and Space Administration's Earth Science Division, through the Cryosphere Sciences Program. R.K. is supported by the Jet Propulsion Laboratory, California Institute of Technology, under contract with NASA.

## Author contributions

M.W. developed analytical tools, processed and analyzed data and results, wrote the paper, and oversaw the project. C.P. assisted with theoretical development of cyclone methodology, designed and performed statistical analyses, and made substantial contributions to the manuscript. L.B. conducted inter-comparison analyses and contributed to the paper. R.K. produced the snow depth reconstructions and contributed to the manuscript.

## Competing interests

The authors declare no competing interests.
