## [Peer Review File · Nature Communications]

Reviewers' comments:

Reviewer #1 (Remarks to the Author):

This research is a great example of taking a systems approach to understanding Arctic climate. The authors are exploring the relationship between storm systems and snow accumulation on sea ice, combining methods from both sea ice literature and cyclone tracking literature. I have not seen this sort of combination of methods before, and it's only with such a combination that the research questions can be adequately addressed. Those methods also provide a rich and complex dataset, where both the cause (storms) and effect (snow accumulation on sea ice) have independent Lagrangian components to consider. The strongest thread I saw throughout the results and conclusions is the difference between cyclone-snow relations in the Atlantic and Pacific sectors of the Arctic sea ice regime. Cyclones are more frequent and more intense in the Atlantic sector in winter, and that translates to a greater influence on the accumulation of snow on sea ice. However, there is substantial local variety beyond the general, as well.

I am impressed by the datasets that the authors have been able to bring together, and the general interpretations drawn from the figures seem logically consistent. I also think the idea itself is a strong point. Snow is an important component of the sea ice seasonal cycle, and understanding the role of synoptic scale cyclones is definitely valuable for any efforts to predict seasonal development of that sea ice pack. There are a few threads that could be continued from this work, including: a) improving our ability to ascribe snow accumulation to individual storms in an automated manner, b) deeper investigation into the non-cyclone factors that also contribute to snow accumulation (to isolate causal factors), and c) richer study into the time series of these variables, particularly at the role of climate change. All of these threads presumably should recombine in the future into actionable modifications to climate models that can better predict Arctic Ocean conditions on seasonal time scales.

There are two things I would like to note that are bigger comments; each might be resolved in multiple ways.

1) The statistical results

I think the biggest realm for improvement in the paper right now is the handling of the statistical tests. The tables in the supplemental material are useful, but the words "table" and "regression" both seem to be absent from the main text. It's likely the average reader would never know they exist, and exactly where the "explained variance" came from would be obscure, too. The type of regression being performed is also worth reconsidering. The authors state twice that they are dealing with non-linear relationships (e.g. Line 279), but they say they used linear regression in the supplemental material. That seems to conflict, and it may mean some biases in the R2 values being reported. Secondly, the regressions with multiple x-variables may suffer from multicollinearity since parameters like size, intensity, and frequency of cyclones are often correlated with each other (i.e., not actually independent). Again, that might bias the R2 values. To improve this part, I'd suggest the authors:

- a) Make their statistical methods explicit in the methods section.
- b) Address the questions about non-linearity and multicollinearity. If they're not a problem here, great, but if they are, then modifications to the method are likely needed. For example, I'd be less concerned about considering both the sea-ice freeze-up timing and the storm activity together in one multiple regression.
- c) Be more clear about the supplemental tables existing and even consider just putting one/some in the main text. The statistics are mentioned more than several of the figures; they seem to be a very prominent part of the interpretations. Therefore, I think it makes sense to place them more prominently in the manuscript.

2) The Biases in the Cyclone-Associated Snowfall Calculations

- a) I think the authors do a good job describing biases in the methods section (e.g., by comparing

to satellite data, but I feel in the results and discussion they consistently imply that any unexplained variance is from “alternate mechanisms” without giving biases much screen time. One component is the positive bias in snowfall/precipitation in the reanalyses. That might have some impact, especially if it’s systematic, e.g., more likely to some regions or more likely caused to manifest when cyclones are present or absent.

Related to all that... I’m guessing the authors know the recent Bosivert et al. (2018) paper looking at precipitation biases in reanalyses over the Arctic Ocean because there’s some author overlap. One of the findings in that paper was more frequent trace precipitation events in reanalyses compared to buoys, including for ERA-Interim (although ERA-Interim was much less problematic than other reanalyses). The positive bias in ERA-Interim compared to the satellites may be in part due to excessive trace precipitation, but such a bias is less likely to be at play for cyclone-associated precipitation. In other words, there’s reason to believe that because of the trace precipitation, there might be an under-estimate in how much precipitation is from cyclones since the non-cyclone precipitation is more likely to be spurious. Depending on how important the trace precipitation is to the snow accumulation reconstructions, that might have a small impact on the results. I’m sure the authors are more knowledgeable of the details, but it seems relevant.

b) Another bias discussed is also shown in Figure 12 -- that the comma shape of the cyclone being detected is not fully encompassed even with the buffered radius, so there is an underestimate of cyclone associated snowfall in those grid cells at the lower part of the comma. Such bias is probably a bit less likely to occur over sea ice in the Atlantic sector because the tail-end of the comma is typically dropping to the south over open water. That might have some bearing on the relationship with snow in the Atlantic v. Pacific sectors.

It may be worthwhile to consider how bias would change if the cyclone-associated precipitation method used by Finnis et al. (2009) and Stroeve et al. (2011) were used, which first defines precipitation regions, and then associates those regions with cyclones based on the intersect of the cyclone area and the precipitation region. That method better captures asymmetrical situations. However, I’m not confident it would lead to substantially different results. Therefore, although I think it might be worth testing the Finnis method on a few months of data to really learn the difference, I think the authors could also improve handling of bias simply by enriching the discussion section.

(Sorry, that got long.)

Some line-by-line comments, which are mostly just more specific places to relate to the main comments above:

Line 163-165: I feel like explaining the Arctic Ocean region’s 61% as “it’s mostly seasonal ice” is a little misleading, because I’d wager every region has more seasonal ice than the Lincoln Sea and several regions have more seasonal ice than the Arctic Ocean region. Therefore, it’s more interesting to think about why the Arctic Ocean has a higher explained variance from freeze-up time than peripheral seas like the Kara or Laptev. Also, the Arctic Ocean region stretches from Atlantic Side to Pacific Side. Since the Atlantic and Pacific sides are shown here to have distinct patterns, it wouldn’t be surprising to me if they muddle each other in the Arctic Ocean region, contributing less to regional variance.

Line 238: “Cyclone density” is used here, but that term usually indicates a different measure than the “cyclone event frequency” that the authors are using.

Line 239: The OLS regression is being employed with units of analysis as years for two regions, but then an example is used comparing the averages of two regions to contradict the statistics. Since the units of analysis appear to be different, I don’t think these numbers are really that comparable. It’s perfectly conceivable that there is a link between interannual variability of cyclone frequency and snow depth change without there being a strong link in spatial variability. This

would be a good place to be more verbose about the statistical tests – and perhaps shift a supplemental table to the main show. If the authors don't want to dig deeper, though, then I'd suggest cut the part in parentheses here and keep the comparisons at the same units of analysis.

Line 246: For parallelism, I'd modify that to "greater proportion of the snowfall than cyclones in the Chukchi Sea in 1979-2016.

Line 249: Perhaps the authors mean "seasonal cycle" instead of "seasonal cyclone"?

Cyclone frequency v. intensity section:

Line 252-254: This line in particular, but the section in general as well, was quite confusing until I realized there were tables of statistics in the supplemental section. At the very least, I think a reference to the supplemental tables is needed, but I think more detailed discussion for be even better.

Line 264: The introduction (suddenly, for the reader) of the multiple regression was also confusing. The authors also jump around from depth to vorticity to size in measuring cyclones here, and it can be hard to follow. Again, I think a lot of this gets more apparent with implementing the suggestion from Major comment #1.

Line 287 (and elsewhere): The phrase "frontal snowfall" appears frequently as a mechanism independent of cyclones that can lead to snow accumulation; however since fronts are often a component of extratropical cyclones, the phrase "frontal snowfall" has a lot of overlap with the idea of "cyclone-associated snowfall" (and some snow falling along fronts is captured by the radius method). Therefore, more specific language might be merited early on in the manuscript before using "frontal snowfall" as shorthand.

Line 319: What is the threshold (alpha) being used for statistical significance? I found it in the supplemental tables, but especially since both $p < 0.05$ and $p < 0.10$ are used, I think the authors should be more explicit and/or switch to using the same threshold throughout.

Methods

Line 457: I feel as those reference 40 is an important citation of this statement, and more important than reference 24.

Line 461-462: This sentence is not entirely clear to me, and so the "novelty" of the approach is not clear. If a cyclone back-tracks to the same location as what? Semi-Lagrangian metrics like cyclone center density (e.g., Neu et al. 2013) and cyclone area frequency (e.g., Wernli and Schwierz 2006) have been used in the past, although less frequently than track density.

Line 476: Repeated word "that"

Line 480-481: How precipitation is associated with each cyclone is not 100% clear until looking at Figure 11. After Figure 11 it's clear that only grid cells falling within the buffered radius of the cyclone center are included as cyclone-associated precipitation. However, it isn't clear if the precipitation data were also re-gridded first to match the polar stereographic grid of the SLP fields or if they were left in their native grid.

Line 505: Is the snow depth for a given season (e.g., SON 2010) the average seasonal reconstructed value? Or the maximum depth in that season (i.e., an end of season value)?

Finally, I might have just missed it, but it's currently not clear to me how grid cells that have some years with sea ice cover and some years without are treated in the trend analyses. (There are generally more grid cells with sea ice cover in the winter in the 1980s than the 2010s.)

Figures

Figure 2 caption: Because of the variety of ways that "cyclone frequency" is presented in the

literature, I would suggest making it clear that the “count” in Figure 2 A-C is an area measure (i.e., the number of times a grid cell is encompassed by a cyclone area) rather than a point density measure (i.e., the number of cyclone centers detected within a given area) or line density measure (i.e., number of tracks that move through a given area) in the Figure 2 caption. Changing the legend from “count” to “event count” might also help.

Figure 3: I assume there’s a reason to flip the color bar from dark to white in A-C to white to dark in D-F, but I can see it. If there isn’t a clear reason, I’d make them consistent with each other because they’re both presenting less snow to more snow.

Figure 4 & 7: Similar to figure 2, specifying that it’s an “event count” (rather than a center count) would be helpful.

Figure 5: Here the authors use “cyclone density”, when I believe it should still be described as an event count.

References from this review

Boisvert, L.N., M.A. Webster, A.A. Petty, T. Markus, D.H. Bromwich, and R.I. Cullather, 2018: Intercomparison of Precipitation Estimates over the Arctic Ocean and Its Peripheral Seas from Reanalyses. *J. Climate*, 31, 8441–8462, <https://doi.org/10.1175/JCLI-D-18-0125.1>

Finnis, J., Cassano, J. J., Holland, M. M., Serreze, M. C., & Uotila, P. (2009). Synoptically forced hydroclimatology of major Arctic watersheds in general circulation models; Part 1: the Mackenzie River Basin. *International Journal of Climatology*, 29(9), 1226–1243. <http://doi.org/10.1002/joc.1753>

Neu, U., Akperov, M. G., Bellenbaum, N., Benestad, R., Blender, R., Caballero, R., et al. (2013). IMILAST: A community effort to intercompare extratropical cyclone detection and tracking algorithms. *Bulletin of the American Meteorological Society*, 94(4), 529–547. <http://doi.org/10.1175/BAMS-D-11-00154.2>

Stroeve, J. C., Serreze, M. C., Barrett, A., & Kindig, D. N. (2011). Attribution of recent changes in autumn cyclone associated precipitation in the Arctic. *Tellus A*, 63(4), 653–663. <http://doi.org/10.1111/j.1600-0870.2011.00515.x>

Wernli, H., & Schwierz, C. (2006). Surface cyclones in the ERA-40 dataset (1958-2001). Part I: Novel identification method and global climatology. *Journal of the Atmospheric Sciences*, 63(10), 2486–2507.

Reviewer #2 (Remarks to the Author):

Review of the manuscript entitled “The role of cyclone activity in snow accumulation on Arctic sea ice” by Webster et al.

This manuscript presents very interesting results of the reconstructed snow data on Arctic sea ice and relationships between cyclone characteristics and snow cover properties. Due to a lack of observations, it has remained unclear how snow is distributed spatially on the Arctic sea ice and how it varies with time. In particular, snow influences sea ice energy budgets due to its insulation and high reflectivity nature, and sea ice mass balance through changes in energy budgets and snow-ice conversion. This study is therefore an important contribution to Arctic climate science. I would like to recommend a revision before it is accepted for publication.

General comments:

The manuscript demonstrates interesting and important results. However, it is lengthy with many duplicated discussions and descriptions, which would reduce the visibility of new findings. For example, three seasons have been used in the cyclone and snow data analyses throughout the major parts of the manuscript. The section of "seasonality" can be well merged into other sections. Also, insulating effect of snow on sea ice has been discussed many times throughout the manuscript. The authors may consider revising the related texts to make the paper more concise. By the way, the authors mentioned a number of times of "Section ##". Actually I cannot find the indicated section. I guess that the authors perhaps forgot removing them when revising this manuscript from its earlier version. So, I suggest the authors to double check on this.

Specific comments:

1. Line 47: I would suggest adding "interacts with the seasonal cycle of surface energy budgets to" before "governs".
2. Line 49-50: Snow cannot accumulate on ocean. I suggest revising the sentence of "... reducing ... ocean" to "reducing the heat loss from sea ice, which is overlaid on warm ocean, to the cooler atmosphere".
3. Line 65: I would suggest adding "though the role of cyclones in the increasing poleward moisture transport into the Arctic has been analyzed recently (Villamil-Otero et al., 2018)" after "models and observations".
Villamil-Otero, G.A., Zhang, J., He, J. et al. Role of extratropical cyclones in the recently observed increase in poleward moisture transport into the Arctic Ocean *Adv. Atmos. Sci.* (2018) 35: 85.
<https://doi.org/10.1007/s00376-017-7116-0>
4. Line 76: It would be useful to add definitions of "variance" and "monthly buildup" here.
5. Line 85-86: The authors mentioned other mechanisms causing snowfall in these two lines and throughout the manuscript. I think the definition of cyclone radius may also cause some uncertainties. This is because the precipitation associated with cyclones is not symmetric about the cyclone center. The authors may add a discussion about this.
6. Line 102 and other following up texts describe strong cyclones into the E. Greenland Sea. I believe this would be due to the definition of the E. Greenland Sea. From many other cyclone analyses and fundamental theory, there should not be many strong cyclones traveling to the coastal areas of Greenland. I would suggest the authors to check on this.
7. Line 109-110: The great snow depth in the E. Greenland Sea could be largely caused by convergence of sea ice/snow due to the transpolar drift, because precipitation does not show maximum values along the E. Greenland Sea coast. Please refer to Timo et al. (2016)
Timo et al., 2016: The atmospheric role in the Arctic water cycle: A review on processes, past and future changes, and their impacts, *J. Geophys. Res. Biogeosci.*, 121, 586–620,
[doi:10.1002/2015JG003132](https://doi.org/10.1002/2015JG003132).
8. Line 119-121: This manuscript does not include analysis of moisture advection by cyclones. The authors may cite the paper by Villamil-Otero et al. (2018) list above about the discussion here.
9. Line 130-132: The large poleward moisture transport in the North Atlantic is also another contributor, seeing the paper by Villamil-Otero et al. (2017) above.
10. Line 144-145: This could be also because the variability of snowfall caused by other factors is small.
11. Line 145-148: There could be uncertainties in the statement in this sentence. This is because (1) seeing the comment on line 144-145 above; and (2) most of snow has fallen before the cyclone enters the interior Arctic (i.e., no water to support snowfall in the interior Arctic Ocean).
12. Line 164: In the previous sentence, for the Arctic average, freeze-up date can explain ~20%. But the number changes to from ~61% in the Arctic Ocean. I would suggest the authors to clarify.
13. Line 168-169: Snow accumulation is defined the difference between the first and last date of this period. Snow/ice melt unlikely occur in March. If there is an early melt in May, why is there is zero snow accumulation? This may need a clarification.
14. Line 225-226: If snow depth is used as a measure to get a composite of cyclone frequency,

intensity, and snowfall, I would like the author to provide a brief description how the composites were derived considering that snow depth reaches its criterion on different date at different grid points.

15. Line 242-244: The Lincoln Sea is covered by stable thick ice. Generally there is no clear dynamic and thermodynamic forcing for cyclone intensification there. It would be interesting and useful to discuss why cyclone intensity and snowfall can be larger in the Lincoln Sea than in the Chukchi Sea.

16. Line 255-257: This sentence may be revised to: The relatively higher correlation with the cyclone vorticity is because vorticity is a measure of strength of rotation and, in turn, convergence/uplift, which transport moisture and heat from open water surface into the atmosphere. The moisture can then precipitate as snow onto sea ice.

17. Line 283-284: Please refer to the paper by Villamil-Otero et al. (2017) mentioned above.

18. Line 319: The text below and the figure caption indicate there is no statistically significant trend. I suggest the authors to clarify what "exception" means here.

19. Line 321-324: This is consistent with the recent finding of Beaufort high strengthening and autumn surface wind increase over the Beaufort and Chukchi seas:

Moore 2012: Decadal variability and a recent amplification of the summer Beaufort Sea High. *Geophys. Res. Lett.*, 39, L10807, doi:<https://doi.org/10.1029/2012GL051570>.

Wu, Q.; Zhang, J.; Zhang, X.; Tao, W. Interannual variability and long-term changes of atmospheric circulation over the Chukchi and Beaufort seas. *J. Clim.* 2014, 27, 4871–4889.

Stegall S T and Zhang J 2012 Wind field climatology, changes, and extremes in the Chukchi–Beaufort seas and Alaska north slope during 1979–2009 *J. Clim.* 25 8075–89.

Zhang, J., S. T. Stegall, and X. Zhang, 2018: Wind–sea surface temperature–sea ice relationship in the Chukchi–Beaufort Seas during autumn. *Environ. Res. Lett.*, 13, 034008, <https://doi.org/10.1088/1748-9326/aa9adb>.

20. Line 326-328: Decrease in cyclone-induced snowfall can also cause a decrease in its contribution even if the snowfall by other systems does not change. The decrease in cyclone-induced snowfall is consistent with the decrease in cyclone frequency and intensity, as well as strengthening of the Beaufort high as mentioned above. Considering this, I suggest to revise the next sentence, because there is no evidence of other system-caused snowfall increase in current study.

21. Line 338: I would suggest revising "has caused" to "may have greatly caused", because of effect of sea ice convergence to this area.

22. Line 752: Figure 9 should be Figure 10.

Reviewer #3 (Remarks to the Author):

Recommendation: major revision

Review by Paul Kushner, University of Toronto

General comments

1. In this study, the authors investigate controls exerted by cyclones on snow cover on Arctic sea ice. They simultaneously characterize Arctic cyclone tracks and activity, their related snowfall characteristics, and snow depth, examining climatology, climate variability, and trends of these quantities. Data sources include reanalysis, satellite derived snowfall, ice mass balance/motion/concentration data. The authors quantify Arctic-cyclone contributions to total snowfall and infer contributions to snow depth, across regions and seasons of the Arctic. Analysis is also devoted to understanding sensitivities to different characteristics of cyclones, including frequency and depth. Cyclone activity explains about half and sea ice freeze up timing explains about a fifth of the variance in snow depth, and strong regional and seasonal variations across the Arctic are documented. The findings are characterized as heterogeneous and complex, and are taken as a starting point for better understanding of the mechanisms of snow on sea ice.

2. This ambitious and original study deals with a cutting edge topic in Arctic and global climate

science. As articulated in this study and in other publications (including several from led by this author team), the characteristics of snow on sea ice controls important sea ice characteristics and provides a major observational constraint for retrieval of sea ice thickness from available satellite observations. The contributions of extratropical cyclones to Arctic snow on sea ice has not previously been studied, to my knowledge, in such a comprehensive manner. The study makes clear the challenge of characterizing Arctic cyclones and then quantitatively elucidating their contribution to snowfall on sea ice. It broadly provides a very strong basis for future work, and advances understanding of this tricky area considerably. Key figures provide an excellent visualization of the patterns and complexities: Fig. 2 showing the Arctic cyclone and snowfall characteristics, Fig. 3 showing the snow depth reconstruction, Figs. 6-7 giving an overview of the seasonal evolution of the snow-on-sea-ice impact, and Fig. 8 showing revealing case studies. The background for the paper in terms of the previous literature is excellent, the write up generally clear, and evidence provided for key conclusions is generally persuasive.

Characterizing Arctic cyclone impact on snow on sea ice is an interdisciplinary topic requiring input from both atmospheric and cryospheric science as well as a mastery of multiple observational datasets. Given this and the importance of the topic, the material written up in this study would be of interest and value to the atmosphere, cryosphere, and climate science community.

3. That being said, the current manuscript has weaknesses that lead to my recommendation of "major revision": the study is a bit too long (does it satisfy Nature Communications word limits?), the write up is repetitive and loose, there are mistakes in the document preparation, and some conclusions are only weakly supported by the presented evidence. Fig. 3 is potentially excellent but currently confusing, Fig. 4 is hardly referred to at all, and there is too much space devoted to the statistically insignificant trends found in Figs. 9 and 10 (which is captioned as Fig. 9). The material in supplementary information is not well documented and the use of this material is unclear. Sleuthing skills are required to make it through the current manuscript. With the exception of the trend analysis, these concerns can be addressed without changing the important conclusions, and the recommendations here are intended to increase the impact of the work.

4. The statistical analysis is not well explained and leaves unopened questions.

The authors need to be more explicit about the physical-statistic models they have in mind with all the possible control variables. The statistical analysis starts sneaking in at l.163, where it is claimed that 20% of the variance comes from freeze-up date. Further digging shows that this is found in the supplementary information, Table S2.

Similarly, e.g. at l.252 statements related to ranges variance explained come from the tables in the supplementary information - these should be referenced.

There is insufficient explicit recognition in the text that several of the processes/mechanisms studied could be cross correlated, including interannual variability in cyclone frequency and intensity, cyclone activity and sea-ice freeze up dates, etc., which could affect interpretation. This is hinted at starting l.267 but not really stated explicitly.

There is no explanation of how r^2 of as small as 0.03 ($r \sim 0.17$) could be statistical significant (e.g. Table S3). I assume the authors are assuming zero serial correlation for a 38 year time series, but is this correct, i.e. are all years statistically independent? There is no explanation of the trend analysis carried out and whether it is assuming linear trends in all fields, which might be naive.

5. 'Interdecadal changes and variability': A lot of space is devoted (l.315-366) to a set of trends maps in Fig. 9 that are not statistically significant - this section is not persuasive. It is worth noting but perhaps not surprising that these noisy patterns end up having consistent signals across variables, but that doesn't make them more significant as temporal signals. A better starting point is Figure 10 (mis-captioned as Fig 9 at l.152) which shows the very interesting pattern of increasing and decreasing snow depth. At least here there are large regions with >90% significance. But here again, trends are emphasized that are not significant (l.340). Perhaps

instead of the seasonal breakdown shown in Fig. 9 the authors could ask if there are trends in accumulated snowfall over the full snow season to help explain Fig. 9. Would spatial aggregation over the different regions help, instead of maps of local trends? E.g. to really cherry pick the authors could calculate area-mean snowfall/cyclone trends at those locations where the snow depth trends are significant.

6. For a contribution to the high impact literature, the material lacks punch and could be tightened. E.g. lines 24-29 in the abstract mirror the first couple of paragraphs in the introduction, some sentences seem to be repeated almost word for word. Also, I wonder why the authors provide the summary in the abstract and again in the introduction (lines 76-94), instead of presenting evidence and using a concluding section to highlight key points before discussing them. Another example: the sea ice freeze up point is introduced twice, once starting at line 157 and again starting at line 271.

7. Has this work been prepared for some other resource or reference? There are erroneous Section references to things like "Section 3.2" and "Section 4.4.2" at lines 117, 286, 362, 478, 484.

8. The peak in cyclonic activity in autumn on the Pacific side (starting at l.173) is hard to see in Fig. 2. Since this is listed as a key conclusion (l.79) more evidence should be provided. Could it be supplemented with the full seasonal cycle of cyclonic activity as a time series, including summer, for both Pacific and Atlantic sectors, eg as a supplementary figure? This would provide context related to the documented summer cyclone maximum from Serreze and Barrett 2008.

9. Points about the figures:

Figure 3: It is not clear why the shading scale is reversed in the two last columns of Fig. 3. In all panels in Fig. 3, is the region south west of the Bering Strait being plotted properly? It is not clear how a snow accumulation of zero at the edges of e.g. Figure 3F and can be accompanied by maxima of Std Dev of accumulation in e.g. Fig. 3I. Can accumulation have a large standard deviation if it has a zero mean? The related discussion at lines 168-171 is confusing. Perhaps mask with a distinctive masking color regions in 3D-F that are near zero in 3G-I.

Fig 4 is mentioned in passing once and not brought up again. This could be replaced by a figure that shows the seasonal cycle in time series form as suggested above.

Fig 9 is stated as having only not statistically significant trends, but at lines 319 and 333 there are significant trends discussed. Not sure a localized trend would qualify as 'field significant'. In any case, are there significant trends anywhere in Fig. 9?

Figures with maps use several different conventions for describing panels and columns. E.g. 'Column 1' versus 'first column' versus no reference to columns. The authors should choose one convention.

10. At a couple of points, e.g. l.36, the contrast in regionality of sea-ice freeze up controls and cyclone activity is emphasized. Cyclone activity r^2 range from 10% (Barents) to 83% (Lincoln) according to Table S3 last column, whereas sea-ice freeze up range from 11% (Barents) to 61% (Arctic Ocean), which is less of a range than cyclone activity, according to Table S2. So looking at these tables it seems there's as much heterogeneity from cyclone activity as from sea-ice freeze up.

11. The idea of snow carried on exported sea ice appears at l.288 as a possible mechanism, where it is not listed previously to this. Why does the ice have to come from 'more northerly latitudes'?

12. The story could be made more interesting by starting with something like Figure 8 which shows contrasting behaviour of snow build up in two regions, also highlighting the excellent use of IMB's here, and then generalizing to the other figures. This is just a suggestion, it's up to the authors whether to pursue it.

Minor comments

1. l.24: complex*,*

2. l.36: 'cyclone activity explained' is vague

3. l.37: overlaps with lines 31-34
4. 'Other snowfall mechanisms' phrase and related text is repeated too many times in the ms, e.g. l.85, l.116, l.362, l.381, etc.
5. l.46: introduction repeats too much of intro material
6. l.63: previous *work has not*
7. l.65: use of word 'models' suggests that authors will look at models here.
8. l.78: delete 'bimodal' - unsuitable term. Suggest: '... regional differences and, in particular, cyclone activity peaks in winter in the Atlantic sector and peaks in autumn in the Pacific sector.' (Is this conclusion really original, given the previous literature?)
9. l.135 forward: the variation in quantities with the ice margin variability is not fully evident in the figure.
10. l.161: 'mute' - do the authors mean 'overwhelm' or 'attenuate'?
11. l.195: delete 'in this region'
12. l.232: cyclone 'density' used here and elsewhere but never defined
13. l.249: seasonal 'cycle'
14. l.310: should be Figure 7 and *8*?
15. l.326: This discusses a marginal feature - should it be kept? In any case the wording is awkward. Suggest 'there is a decrease that is much smaller in magnitude or a slight increase over the ...'
16. l.384: explain up to ~83% . . . -> explain, regionally, over 80%
17. l.401: obscure connection to Fig. S1
18. l.502: No initial conditions -> Initial values of zero
19. l.504: black->blue
20. l.696: 'the reconstructed snow depth' - not clear what this is, refer to methods.
21. l.748: does this sentence require revision given comment above?

We thank the reviewers for their time and energy in reviewing our work and providing constructive feedback. We're deeply appreciative of the suggestions for improving the analysis and manuscript.

Reviewers' comments:

Reviewer #1 (Remarks to the Author):

This research is a great example of taking a systems approach to understanding Arctic climate. The authors are exploring the relationship between storm systems and snow accumulation on sea ice, combining methods from both sea ice literature and cyclone tracking literature. I have not seen this sort of combination of methods before, and it's only with such a combination that the research questions can be adequately addressed. Those methods also provide a rich and complex dataset, where both the cause (storms) and effect (snow accumulation on sea ice) have independent Lagrangian components to consider. The strongest thread I saw throughout the results and conclusions is the difference between cyclone-snow relations in the Atlantic and Pacific sectors of the Arctic sea ice regime. Cyclones are more frequent and more intense in the Atlantic sector in winter, and that translates to a greater influence on the accumulation of snow on sea ice. However, there is substantial local variety beyond the general, as well.

I am impressed by the datasets that the authors have been able to bring together, and the general interpretations drawn from the figures seem logically consistent. I also think the idea itself is a strong point. Snow is an important component of the sea ice seasonal cycle, and understanding the role of synoptic scale cyclones is definitely valuable for any efforts to predict seasonal development of that sea ice pack. There are a few threads that could be continued from this work, including: a) improving our ability to ascribe snow accumulation to individual storms in an automated manner, b) deeper investigation into the non-cyclone factors that also contribute to snow accumulation (to isolate causal factors), and c) richer study into the time series of these variables, particularly at the role of climate change. All of these threads presumably should recombine in the future into actionable modifications to climate models that can better predict Arctic Ocean conditions on seasonal time scales.

Thank you for such a kind and informative summary of our analysis. We feel there are several exciting directions for the community to build on from here, especially those that you've described in (a), (b), and (c).

There are two things I would like to note that are bigger comments; each might be resolved in multiple ways.

1) The statistical results

I think the biggest realm for improvement in the paper right now is the handling of the statistical tests. The tables in the supplemental material are useful, but the words "table" and "regression" both seem to be absent from the main text. It's likely the average reader would never know they exist, and exactly where the "explained variance" came from would be obscure, too. The type of regression being performed is also worth reconsidering. The authors state twice that they are

dealing with non-linear relationships (e.g. Line 279), but they say they used linear regression in the supplemental material. That seems to conflict, and it may mean some biases in the R2 values being reported. Secondly, the regressions with multiple x-variables may suffer from multicollinearity since parameters like size, intensity, and frequency of cyclones are often correlated with each other (i.e., not actually independent). Again, that might bias the R2 values. To improve this part, I'd suggest the authors:

a) Make their statistical methods explicit in the methods section.

Thank you for the helpful suggestion. We added a new section in the methods describing the statistical analysis and also added more details in supplementary materials:

“Statistical Approach

To evaluate the relationship between cyclones and snow accumulation on sea ice, we used univariate linear regressions between the monthly snow accumulation (dependent) and monthly cyclone characteristics (independent) over sea ice. To validate the choice of a linear regression model, we examined linear, cubic, and exponential regressions for all variables across all regions. Linear regressions yielded the lowest root-mean-squared (RMS) errors for the majority of cases (see Supplementary Information), demonstrating the strongest fit among variables. For consistency across variables and regions, we deferred to using a linear regression model. The analysis includes all variables in September-May for 1979-2016, has a sample size of 336, and here and for all other statistical analyses conducted, 0.05 is used for statistical significance. A monthly temporal resolution was chosen to avoid the high-frequency fluctuations in daily snow depth due to ice motion. The cyclone characteristics are calculated as the monthly regional mean of the sum of: (a) count; (b) vorticity; (c) size; and (d) snowfall over sea ice. The monthly sum is taken from the last day of the preceding month to the penultimate day of the relevant month. This one-day offset accounts for the lag of snow accumulation on sea ice following cyclone events.

Using monthly variables over sea ice, we further evaluated the combined effect of cyclone characteristics on snow in a multivariate framework using a sample size of 336. The multivariate models use monthly snow accumulation (dependent) and (a) cyclone count and vorticity; and (b) count, vorticity, and size (independent). The models use the regional mean of the temporal sum of cyclone count, but the regional mean of the temporal mean in cyclone vorticity and size to avoid double counting cyclones and erroneously estimating their effect on snow accumulation. Scatterplots in Supplementary Figure 5c,g-h demonstrate that the cyclone variables considered in the multiple regression framework largely do not exhibit collinearity. This suggests that more cyclone events do not equate to stronger or larger cyclones and establishes that these variables can be treated as independents. Cyclone snowfall was not included in the multivariate analysis to avoid multicollinearity (Supplementary Fig. 5a,e)

Given the strong positive correlation between cyclone snowfall and snow accumulation (see Fig. 5b and Supplementary Table 2), we analyzed the relationship between cyclone count and vorticity and cyclone-associated snowfall. Regional averages of the monthly sums in these variables were used, resulting in a sample size of 456. Supplementary Figure 5a,e demonstrates the positive correlations between cyclone snowfall and count and vorticity and a strong fit of the linear regression model.

Prior to the trend analysis, we tested for autocorrelation among variables using the Durbin-Watson Test. The test reveals that, despite a few exceptions, cyclone counts, cyclone

snowfall, vorticity, total snowfall, and total precipitation are not autocorrelated. The exceptions are: cyclone snowfall in the E. Siberian Sea and total precipitation in the Arctic Ocean (autocorrelated); total snowfall in the Arctic Ocean and Laptev Sea and vorticity in the Arctic Ocean (inconclusive). Furthermore, we use zero initial values in the snow depth reconstruction each year, which removes any carryover of snow from the preceding year. Therefore, with a few exceptions, the assumption of a zero serial correlation is justifiable for variables in the 37-year time-series. We examined linear, cubic, and exponential regressions for all variables across regions and chose linear regressions due to the strongest fits for exploring whether discernable trends in variables exist for 1979-2016. We note that, for several variables, the trends were inconclusive. We found no significant or coherent relationships between freeze-up and cyclone characteristics. Collinearity between these variables is unlikely to affect the interpretation of the linear regressions and interdecadal trends.”

In supplementary materials:

“Cyclone variables such as count and snowfall can co-vary (Supplementary Fig. 5a,e). However, the correlations are regionally dependent and it is important to understand the influence of each cyclone variable on snow accumulation in a univariate framework. Our analysis of the data characteristics suggests that cyclone count, vorticity, and size largely do not exhibit collinearity (Supplementary Fig. 5c,g,h). Therefore, in the climatology, more cyclone events do not equate to stronger or larger cyclones and these variables can be considered in a multivariate framework. Prevailing atmospheric conditions and oscillations may influence interannual variability in cyclone characteristics. However, the univariate and multivariate regression analysis does not consider the correlation in cyclone variables over time, but explores the relationships between individual monthly measures of cyclone characteristics and the snow accumulation. While some resulting correlations are small, they are still statistically significant given the large sample size (336 or 456). Nevertheless, we note that some individual cyclone characteristics do not have strong relationships with snow accumulation given the small correlations. For the trend analysis, we explored linear, cubic, and exponential regressions for all variables across regions. Linear regressions exhibited the strongest fits with the data (i.e. lowest root-mean-squared errors) with only two exceptions. The exceptions were total snowfall and total precipitation, which cubic and exponential models yielded the lowest RMS errors, respectively. Given that the scope of the analysis does not focus on trends in total snowfall or total precipitation, we deferred to using linear regressions for all variables in the trend analysis for consistency in approach.”

b) Address the questions about non-linearity and multicollinearity. If they’re not a problem here, great, but if they are, then modifications to the method are likely needed. For example, I’d be less concerned about considering both the sea-ice freeze-up timing and the storm activity together in one multiple regression.

Thank you for raising this issue. They were indeed a problem between cyclone snowfall and snow accumulation in the multivariate regression. We made changes in our multivariate analysis to exclude cyclone snowfall as one of the independent variables and adjusted the findings according to the new results. Regarding linearity vs. non-linearity, we examined linear, cubic, and exponential regressions for all variables across all regions. Linear regressions yielded the lowest root-mean-squared (RMS) errors for the majority of cases, demonstrating the strongest fit among variables. The exceptions were total snowfall and total precipitation, which cubic and exponential models yielded the lowest RMS errors, respectively. We show the linear regressions

in scatterplots as new figures in supplementary information (but shown below). For consistency across variables and regions, we deferred to using a linear regression model. Given that trends in total snowfall and total precipitation were not the central focus of the analysis, we felt this decision was appropriate.

c) Be more clear about the supplemental tables existing and even consider just putting one/some in the main text. The statistics are mentioned more than several of the figures; they seem to be a very prominent part of the interpretations. Therefore, I think it makes sense to place them more prominently in the manuscript.

This is a great idea! Thank you. We've removed Figure 4 and replaced it with a figure (now Figure 5, below) of the results from the univariate and multivariate analysis, as well as the correlations between annual snow accumulation and sea-ice freeze-up. We reference supplementary materials more directly.

2) The Biases in the Cyclone-Associated Snowfall Calculations

a) I think the authors do an good job describing biases in the methods section (e.g., by comparing to satellite data, but I feel in the results and discussion they consistently imply that any unexplained variance is from “alternate mechanisms” without giving biases much screen time. One component is the positive bias in snowfall/precipitation in the reanalyses. That might have some impact, especially if it’s systematic, e.g., more likely to some regions or more likely caused to manifest when cyclones are present or absent.

Related to all that... I’m guessing the authors know the recent Bosivert et al. (2018) paper looking at precipitation biases in reanalyses over the Arctic Ocean because there’s some author overlap. One of the findings in that paper was more frequent trace precipitation events in reanalyses compared to buoys, including for ERA-Interim (although ERA-Interim was much less problematic than other reanalyses). The positive bias in ERA-Interim compared to the satellites may be in part due to excessive trace precipitation, but such a bias is less likely to be at play for cyclone-associated precipitation. In other words, there’s reason to believe that because of the trace precipitation, there might be an under-estimate in how much precipitation is from cyclones since the non-cyclone precipitation is more likely to be spurious. Depending on how important the trace precipitation is to the snow accumulation reconstructions, that might have a small impact on the results. I’m sure the authors are more knowledgeable of the details, but it seems relevant.

That’s an excellent point regarding reanalysis biases. In the main text, we added content on the issue of trace precipitation (and cyclone asymmetry) in the discussion section:

“A key result from our work is that cyclones account for more than 80% of total snowfall in the Atlantic regions. These regions appear to be more reliant on cyclone snowfall to establish a snowpack. Conversely, cyclones account for only ~50% of total snowfall in the Pacific region and therefore snow input from other mechanisms may be equally important for snowpack establishment. There are factors to consider when interpreting this result, specifically regarding “missed” snowfall during cyclone events and systematic trace precipitation. Not all cyclone snowfall may be captured by the spatial buffer in the tracking methods, in particular, the tails of comma-like pattern of cyclone precipitation (Fig. 1b). Equally important, trace precipitation, or “residual drizzle,” is a known issue in reanalysis products, which results in too frequent and too much snowfall outside of true snowfall events (Supplementary Figs. 2 and 3)³⁵⁻³⁶. Thus, both the “missed” cyclone snowfall and trace precipitation may erroneously increase the proportion of snowfall outside of cyclone events and lessen the amount that cyclones truly contribute. The effects of such biases necessitate the collection of process-oriented observations of cloud microphysics and precipitation to improve understanding of precipitation mechanisms and their treatment in models.

To assess ERA-Interim snowfall and precipitation, we compared these variables with those from NASA’s Atmospheric Infrared Sounder (AIRS)³⁷ and CloudSat³⁸ (Supplementary Figs. 2 and 3). Throughout the accumulation season (September-May), ERA-Interim and CloudSat produce similar snowfall, with ERA-Interim producing an average of 1.20 ± 0.08 mm-day⁻¹ over the sea-ice pack (defined as 15% ice concentration and greater) compared to 1.11 ± 0.10 mm-day⁻¹ from CloudSat. When total precipitation from ERA-Interim and AIRS were compared, AIRS produced less precipitation over sea ice, with ERA-Interim having an average of 1.93 ± 0.13 mm-day⁻¹ compared to 1.31 ± 0.10 mm-day⁻¹ from AIRS. In both comparisons,

ERA-Interim produces more snowfall and total precipitation, which underscores the possibility that trace precipitation is over-produced in the reanalysis product and leads to an underestimation of cyclones' contribution to snow accumulation on sea ice."

b) Another bias discussed is also shown in Figure 12 -- that the comma shape of the cyclone being detected is not fully encompassed even with the buffered radius, so there is an underestimate of cyclone associated snowfall in those grid cells at the lower part of the comma. Such bias is probably a bit less likely to occur over sea ice in the Atlantic sector because the tail-end of the comma is typically dropping to the south over open water. That might have some bearing on the relationship with snow in the Atlantic v. Pacific sectors.

It may be worthwhile to consider how bias would change if the cyclone-associated precipitation method used by Finnis et al. (2009) and Stroeve et al. (2011) were used, which first defines precipitation regions, and then associates those regions with cyclones based on the intersect of the cyclone area and the precipitation region. That method better captures asymmetrical situations. However, I'm not confident it would lead to substantially different results. Therefore, although I think it might be worth testing the Finnis method on a few months of data to really learn the difference, I think the authors could also improve handling of bias simply by enriching the discussion section.

Thank you for raising this issue. You have a good point here. We tested out the Finnis/Stroeve method using 6-hourly data from January 2009, based on the description of the methods in Stroeve et al. (2011). We examined native and regrided precipitation data as well. Below is a plot of the average precipitation of cyclones at each time-step using the two methods and native and regrided precipitation. MU stands for the Melbourne University tracker used in our analysis, although we note that we add a 2-degree spatial buffer to their approach to identify cyclone-associated snowfall. The numbers in parentheses are averages of the time-series.

To understand these results further, we looked into the spatial pattern of cyclone precipitation. Below is an example from Jan. 1st 2009 of the different methods identifying cyclone-associated snowfall. The yellow dots represent the area over which precipitation is associated with a cyclone in our methods(b-f). The pink dots represent the distance threshold (250 km) used in the Finnis/Stroeve method in (c) and (e). Already, one can see that this approach doesn't capture all of the cyclone-associated precipitation. In Finnis/Stroeve, an additional threshold is applied to precipitation (1.5 mm day^{-1}) to eliminate any small-magnitude precipitation occurring outside of

the cyclone event. The precipitation that meets this threshold criteria is shown by cyan dots in (d) and (f).

Both approaches have biases. In Finniss/Stroeve, large-magnitude precipitation is preferentially sampled (yielding a high average), but not all of the cyclone precipitation is captured (yielding a low sum). In our approach, we sample all magnitudes of precipitation, which non-cyclone precipitation (yielding a low average), but our approach tends to have better coverage of the cyclone precipitation (yielding a high sum). There is much more work to be done on disentangling the effects of different biases from different methods, and it certainly warrants a separate paper in of itself. We expand the discussion on the possible effects from biases in our approach in the manuscript.

(Sorry, that got long.) Thank you! Your feedback was extremely helpful, and we greatly appreciate your time in reviewing our work.

Some line-by-line comments, which are mostly just more specific places to relate to the main comments above:

Line 163-165: I feel like explaining the Arctic Ocean region’s 61% as “it’s mostly seasonal ice” is a little misleading, because I’d wager every region has more seasonal ice than the Lincoln Sea and several regions have more seasonal ice than the Arctic Ocean region. Therefore, it’s more interesting to think about why the Arctic Ocean has a higher explained variance from freeze-up time than peripheral seas like the Kara or Laptev. Also, the Arctic Ocean region stretches from Atlantic Side to Pacific Side. Since the Atlantic and Pacific sides are shown here to have distinct patterns, it wouldn’t be surprising to me if they muddle each other in the Arctic Ocean region,

contributing less to regional variance.

Yes, you are correct. We revised the content as follows to contrast the relatively higher correlations between the Lincoln Sea and lower correlations in some of the peripheral seas where seasonal ice dominates:

“Compared to some peripheral seas (Kara, Chukchi, E. Greenland), the Lincoln Sea has earlier freeze-up and less seasonal ice, yet it has a stronger correlation between annual snow accumulation and freeze-up. The weaker relationship in some “seasonal-ice” regions may arise from several factors: (a) total snowfall may be proportionally less in autumn than in winter and spring, and thus the annual snow accumulation may be less sensitive to the date of autumnal freeze-up, (b) frequent cyclone events may occur throughout the accumulation season, overwhelming the effect of late freeze-up on snow accumulation, and (c) a later shift in the seasonal cycle of snowfall may result in increased winter snowfall, compensating for the lack of snow accumulation in autumn.”

Line 238: “Cyclone density” is used here, but that term usually indicates a different measure than the “cyclone event frequency” that the authors are using.

We agree and opted to use “count” and “event count” for consistency rather than a mix of count, event count, density, and frequency. We also added a definition based on your description below for clarity.

Line 239: The OLS regression is being employed with units of analysis as years for two regions, but then an example is used comparing the averages of two regions to contradict the statistics. Since the units of analysis appear to be different, I don’t think these numbers are really that comparable. It’s perfectly conceivable that there is a link between interannual variability of cyclone frequency and snow depth change without there being a strong link in spatial variability. This would be a good place to be more verbose about the statistical tests – and perhaps shift a supplemental table to the main show. If the authors don’t want to dig deeper, though, then I’d suggest cut the part in parentheses here and keep the comparisons at the same units of analysis.

We agree that we were mixing two types of results in a confusing manner and revised this section. We added a new section in the methods describing the statistical approach in detail and also included a discussion in supplementary materials. We revised the content in the section you’re referring to (see below) to exclude the univariate and multivariate statistics, saving those results for the following section “Cyclone count vs. intensity”. In the Cyclone count vs. intensity section, we make clear that the results are from the univariate and multivariate analyses. We also followed your suggestion to bring the statistical results into the main text by creating a new figure (Figure 5) based on the tables in supplementary materials. We refer to the new figure, as well as figures and tables in supplementary materials, more directly in the main text.

“These results may appear to suggest that more cyclone events are needed to establish a snow cover that is considerably deeper (e.g. Atlantic sector) (Figs. 4a-c and 7g). However, this relationship varies regionally. While the number of cyclone counts in the Lincoln and Chukchi seas is roughly equal by the date that 50% of their snowpacks are established, the snow is 16% deeper in the Lincoln Sea.”

Line 246: For parallelism, I’d modify that to “greater proportion of the snowfall than cyclones in

the Chukchi Sea in 1979-2016.

Revised; thank you.

Line 249: Perhaps the authors mean “seasonal cycle” instead of “seasonal cyclone”?

Yes, we do. Thank you for spotting this.

Cyclone frequency v. intensity section:

Line 252-254: This line in particular, but the section in general as well, was quite confusing until I realized there were tables of statistics in the supplemental section. At the very least, I think a reference to the supplemental tables is needed, but I think more detailed discussion for be even better.

Thank you for the suggestion. We added an in-depth description of the statistical analysis in the Methods and supplementary materials, directly reference new figures related to the statistical results, and also refer readers to both the Methods and Supplementary Materials in this section.

Line 264: The introduction (suddenly, for the reader) of the multiple regression was also confusing. The authors also jump around from depth to vorticity to size in measuring cyclones here, and it can be hard to follow. Again, I think a lot of this gets more apparent with implementing the suggestion from Major comment #1.

Thank you! We hope the revised content improves this greatly following the changes listed in the previous comment.

Line 287 (and elsewhere): The phrase “frontal snowfall” appears frequently as a mechanism independent of cyclones that can lead to snow accumulation; however since fronts are often a component of extratropical cyclones, the phrase “frontal snowfall” has a lot of overlap with the idea of “cyclone-associated snowfall” (and some snow falling along fronts is captured by the radius method). Therefore, more specific language might be merited early on in the manuscript before using “frontal snowfall” as shorthand.

We agree, and have added more detailed descriptions about other possible snowfall processes rather than using this as an umbrella term incorrectly.

Line 319: What is the threshold (α) being used for statistical significance? I found it in the supplemental tables, but especially since both $p < 0.05$ and $p < 0.10$ are used, I think the authors should be more explicit and/or switch to using the same threshold throughout.

The threshold is now $p < 0.05$ for statistical significance for all univariate, multivariate, and trend analyses. This is now explicit in the methods – thank you for catching this.

Methods

Line 457: I feel as those reference 40 is an important citation of this statement, and more important than reference 24.

We agree.

Line 461-462: This sentence is not entirely clear to me, and so the “novelty” of the approach is not clear. If a cyclone back-tracks to the same location as what? Semi-Lagrangian metrics like cyclone center density (e.g., Neu et al. 2013) and cyclone area frequency (e.g., Wernli and Schwierz 2006) have been used in the past, although less frequently than track density.

You're correct. We've revised this section accordingly and referenced previous works that have used a similar semi-Lagrangian approach.

Line 476: Repeated word "that"

Thank you; fixed.

Line 480-481: How precipitation is associated with each cyclone is not 100% clear until looking at Figure 11. After Figure 11 it's clear that only grid cells falling within the buffered radius of the cyclone center are included as cyclone-associated precipitation. However, it isn't clear if the precipitation data were also re-gridded first to match the polar stereographic grid of the SLP fields or if they were left in their native grid.

We combined this figure with Figure 1 to help illustrate our approach, and we also included a description of how the precipitation fields were treated. They were regridded to match the polar stereographic grid of the SLP fields.

Line 505: Is the snow depth for a given season (e.g., SON 2010) the average seasonal reconstructed value? Or the maximum depth in that season (i.e., an end of season value)?

The snow depth is the average for a given season. The snow accumulation is the last day's snow depth minus the first day's snow depth for a given season. For the spring season, we have to account for spring sea-ice melt at lower latitudes by taking the seasonal maximum snow depth (which typically occurs in early May) minus the first day's snow depth. If we didn't use the maximum snow depth for spring snow accumulation, we'd get negative values of snow accumulation in southern latitudes.

Finally, I might have just missed it, but it's currently not clear to me how grid cells that have some years with sea ice cover and some years without are treated in the trend analyses. (There are generally more grid cells with sea ice cover in the winter in the 1980s than the 2010s.)

Unlike in the univariate and multivariate analysis, we take the entire regional average regardless of ice area. We computed trends twice: one dependent on the ice area (not shown) and one using the regional averages of cyclone variables over time. For the ice area-dependent results, the loss of sea ice dominates the detectable changes in cyclone variables, producing negative results everywhere. Therefore, we chose to evaluate the regional means to eliminate the influence of sea-ice loss. We've clarified this in the text.

Figures

Figure 2 caption: Because of the variety of ways that "cyclone frequency" is presented in the literature, I would suggest making it clear that the "count" in Figure 2 A-C is an area measure (i.e., the number of times a grid cell is encompassed by a cyclone area) rather than a point density measure (i.e., the number of cyclone centers detected within a given area) or line density measure (i.e., number of tracks that move through a given area) in the Figure 2 caption.

Changing the legend from "count" to "event count" might also help.

Thank you for the helpful suggestion. We agree and opted to use "count" and "event count" for consistency. We also added a definition based on your description to avoid any confusion.

Figure 3: I assume there's a reason to flip the color bar from dark to white in A-C to white to dark in D-F, but I can see it. If there isn't a clear reason, I'd make them consistent with each

other because they're both presenting less snow to more snow.

There is no good reason for the color flip, so we changed the color schemes so that the lightest shades represent the highest values. We also changed the colors between variables so that it's more apparent that these are different variables.

Figure 4 & 7: Similar to figure 2, specifying that it's an "event count" (rather than a center count) would be helpful.

We agree; thank you for the suggestion.

Figure 5: Here the authors use "cyclone density", when I believe it should still be described as an event count.

Agreed.

References from this review

Boisvert, L.N., M.A. Webster, A.A. Petty, T. Markus, D.H. Bromwich, and R.I. Cullather, 2018: Intercomparison of Precipitation Estimates over the Arctic Ocean and Its Peripheral Seas from Reanalyses. *J. Climate*, 31, 8441–8462, <https://doi.org/10.1175/JCLI-D-18-0125.1>

Finnis, J., Cassano, J. J., Holland, M. M., Serreze, M. C., & Uotila, P. (2009). Synoptically forced hydroclimatology of major Arctic watersheds in general circulation models; Part 1: the Mackenzie River Basin. *International Journal of Climatology*, 29(9), 1226–1243. <http://doi.org/10.1002/joc.1753>

Neu, U., Akperov, M. G., Bellenbaum, N., Benestad, R., Blender, R., Caballero, R., et al. (2013). IMILAST: A community effort to intercompare extratropical cyclone detection and tracking algorithms. *Bulletin of the American Meteorological Society*, 94(4), 529–547. <http://doi.org/10.1175/BAMS-D-11-00154.2>

Stroeve, J. C., Serreze, M. C., Barrett, A., & Kindig, D. N. (2011). Attribution of recent changes in autumn cyclone associated precipitation in the Arctic. *Tellus A*, 63(4), 653–663. <http://doi.org/10.1111/j.1600-0870.2011.00515.x>

Wernli, H., & Schwierz, C. (2006). Surface cyclones in the ERA-40 dataset (1958-2001). Part I: Novel identification method and global climatology. *Journal of the Atmospheric Sciences*, 63(10), 2486–2507.

Reviewer #2 (Remarks to the Author):

Review of the manuscript entitled "The role of cyclone activity in snow accumulation on Arctic sea ice" by Webster et al.

This manuscript presents very interesting results of the reconstructed snow data on Arctic sea ice and relationships between cyclone characteristics and snow cover properties. Due to a lack of observations, it has remained unclear how snow is distributed spatially on the Arctic sea ice and

how it varies with time. In particular, snow influences sea ice energy budgets due to its insulation and high reflectivity nature, and sea ice mass balance through changes in energy budgets and snow-ice conversion. This study is therefore an important contribution to Arctic climate science. I would like to recommend a revision before it is accepted for publication.

Thank you for the thoughtful summary of our analysis. We appreciate it.

General comments:

The manuscript demonstrates interesting and important results. However, it is lengthy with many duplicated discussions and descriptions, which would reduce the visibility of new findings. For example, three seasons have been used in the cyclone and snow data analyses throughout the major parts of the manuscript. The section of “seasonality” can be well merged into other sections. Also, insulating effect of snow on sea ice has been discussed many times throughout the manuscript. The authors may consider revising the related texts to make the paper more concise. By the way, the authors mentioned a number of times of “Section ##”. Actually I cannot find the indicated section. I guess that the authors perhaps forgot removing them when revising this manuscript from its earlier version. So, I suggest the authors to double check on this.

Thank you for the constructive comments. They were valuable guidance for improving the manuscript. Following your suggestions, we:

- Reduced the length by nearly 2,000 words including the addition of a statistics section in the methods.*
- Tailored the text to emphasize the key findings rather than give a seasonal breakdown of each aspect.*
- Cut out duplicated text, and consolidated and merged material, including merging the “Seasonality” section into other sections.*
- We removed the redundant information on snow’s insulating effects, as well as other aspects of snow that were unnecessarily repeated.*
- We corrected the typos of “Section XX.” We apologize for the sloppiness. Our initial plan was to submit the manuscript to J. Climate, but we were encouraged to first submit it to Nature Communications.*

Specific comments:

1. Line 47: I would suggest adding "interacts with the seasonal cycle of surface energy budgets to" before "governs".

Thank you for the helpful suggestion. We opted to remove this paragraph to cut down on the redundant background information in the abstract and introduction.

2. Line 49-50: Snow cannot accumulate on ocean. I suggest revising the sentence of “... reducing ... ocean” to "reducing the heat loss from sea ice, which is overlaid on warm ocean, to the cooler atmosphere".

Thank you, again, for the helpful suggestion. As mentioned before, we removed this paragraph to streamline the content.

3. Line 65: I would suggest adding “though the role of cyclones in the increasing poleward moisture transport into the Arctic has been analyzed recently (Villamil-Otero et al., 2018)” after

“models and observations”.

Villamil-Otero, G.A., Zhang, J., He, J. et al. Role of extratropical cyclones in the recently observed increase in poleward moisture transport into the Arctic Ocean *Adv. Atmos. Sci.* (2018) 35: 85. <https://doi.org/10.1007/s00376-017-7116-0>

Thank you very much for sharing this analysis with us. It's especially informative for our work. We found it to be a valuable reference several times throughout the revised manuscript. We did not reference it specifically for this sentence, however, because we did not want to give readers the wrong impression that we investigated moisture transport into the Arctic Ocean in this analysis (we didn't, but it would certainly be an exciting follow-up from this and the Villamil-Otero's works).

4. Line 76: It would be useful to add definitions of “variance” and “monthly buildup” here.

We removed the content here to consolidate the key findings in the manuscript. We do follow your suggestions later in the text and provide clearer terminology for describing the results.

5. Line 85-86: The authors mentioned other mechanisms causing snowfall in these two lines and throughout the manuscript. I think the definition of cyclone radius may also cause some uncertainties. This is because the precipitation associated with cyclones is not symmetric about the cyclone center. The authors may add a discussion about this.

That's an excellent point regarding cyclone asymmetry. We expanded the content on this issue in the discussion section:

“A key result from our work is that cyclones account for more than 80% of total snowfall in the Atlantic regions. These regions appear to be more reliant on cyclone snowfall to establish a snowpack. Conversely, cyclones account for only ~50% of total snowfall in the Pacific region and therefore snow input from other mechanisms may be equally important for snowpack establishment. There are factors to consider when interpreting this result, specifically regarding “missed” snowfall during cyclone events and systematic trace precipitation. Not all cyclone snowfall may be captured by the spatial buffer in the tracking methods, in particular, the tails of comma-like pattern of cyclone precipitation (Fig. 1b). Equally important, trace precipitation, or “residual drizzle,” is a known issue in reanalysis products, which results in too frequent and too much snowfall outside of true snowfall events (Supplementary Figs. 2 and 3)35-36. Thus, both the “missed” cyclone snowfall and trace precipitation may erroneously increase the proportion of snowfall outside of cyclone events and lessen the amount that cyclones truly contribute. The effects of such biases necessitate the collection of process-oriented observations of cloud microphysics and precipitation to improve understanding of precipitation mechanisms and their treatment in models.

To assess ERA-Interim snowfall and precipitation, we compared these variables with those from NASA's Atmospheric Infrared Sounder (AIRS)37 and CloudSat38 (Supplementary Figs. 2 and 3). Throughout the accumulation season (September-May), ERA-Interim and CloudSat produce similar snowfall, with ERA-Interim producing an average of 1.20 ± 0.08 mm-day⁻¹ over the sea-ice pack (defined as 15% ice concentration and greater) compared to 1.11 ± 0.10 mm-day⁻¹ from CloudSat. When total precipitation from ERA-Interim and AIRS were compared, AIRS produced less precipitation over sea ice, with ERA-Interim having an average of 1.93 ± 0.13 mm-day⁻¹ compared to 1.31 ± 0.10 mm-day⁻¹ from AIRS. In both comparisons, ERA-Interim produces more snowfall and total precipitation, which underscores the possibility

that trace precipitation is over-produced in the reanalysis product and leads to an underestimation of cyclones' contribution to snow accumulation on sea ice."

6. Line 102 and other following up texts describe strong cyclones into the E. Greenland Sea. I believe this would be due to the definition of the E. Greenland Sea. From many other cyclone analyses and fundamental theory, there should not been many strong cyclones traveling to the costal areas of Greenland. I would suggest the authors to check on this.

Yes, you're correct that it is a matter of definition of the region being analyzed. Our E. Greenland Sea region includes the open ocean portion in the N. Atlantic, as illustrated in Figure 1a. For the univariate and multivariate statistics, we only analyzed values over the ice-covered area in this region (and now state this explicitly in the Methods). Averages over the sea-ice areas still show that relatively stronger cyclones transit the sea ice area in this region relative to cyclones in other regions. For the trend analysis, we include the entire region's values. Otherwise, the loss of Arctic sea ice would dominate any detectable long-term change in cyclone activity (i.e. causing erroneously negative trends in cyclone variables). We clarified this in the revision. Double-checking the distributions of cyclone counts and intensities for this and other regions, our numbers are consistent with the literature and well within the values between different cyclone trackers and reanalysis products.

7. Line 109-110: The great snow depth in the E. Greenland Sea could be largely caused by convergence of sea ice/snow due to the transpolar drift, because precipitation does not show maximum values along the E. Greenland Sea coast. Please refer to Timo et al. (2016)

Timo et al., 2016: The atmospheric role in the Arctic water cycle: A review on processes, past and future changes, and their impacts, *J. Geophys. Res. Biogeosci.*, 121, 586–620, doi:10.1002/2015JG003132.

Thank you for the valuable reference; we found it extremely insightful and included it as a reference. We agree that there are several factors that may contribute to the large snow depths in this region. For the modeled snow depths in this analysis, we do not account for ice convergence, meaning that the snow depth does not increase during a convergent event. All results from various reconstructed snow depth approaches show the deepest snowpack is found in this region (Kwok and Cunningham, 2008; Blanchard-Wrigglesworth et al., 2018; Petty et al., 2018; Webster et al., 2018), regardless of the reanalysis snowfall product used, treatment of ice convergence, or ice motion product used. Thus, convergence does not explain the deep snow in the E. Greenland Sea. Interestingly, the (admittedly sparse) observations from this region suggest three things: 1) the snow depth in this region is the deepest observed in the Arctic based on a North Pole drifting ice station, a U.S. Navy ice camp, and the N-ICE cruise measurements; 2) Blowing snow lost to leads may be an important mechanism here. Observations from the N-ICE campaign in 2015 showed zero change in mean snow depth after major snowfall events, which also coincided with strong winds; they attributed this net-zero change due to the loss of blowing snow to leads, which were prominent features throughout the local area. This may suggest that modeled snow depths should include a sink term of snow lost to leads, which may be more important in areas where ice motion is high, sea ice cover is fractured and/or the sea ice concentration is low, like in the E. Greenland Sea; and 3) Flooded snow or snow-ice formation, an unusual phenomenon for snow on Arctic sea ice, was observed during two separate field campaigns in this region. This may be another "sink" that is missing from the model, leading to deeper snow. Lastly, when looking at maps of snowfall distributions from ERA-Interim where sea

ice is present, this region receives some of the largest snowfall relative to other sea ice-covered regions. If multiyear ice is advected into Fram Strait and the E. Greenland Sea, there is ample time for heavy and frequent snowfall (see figure below) to accumulate.

8. Line 119-121: This manuscript does not include analysis of moisture advection by cyclones. The authors may cite the paper by Villamil-Otero et al. (2018) list above about the discussion here.

We deleted this description in the revision, but addressed this point in the following comment.

9. Line 130-132: The large poleward moisture transport in the North Atlantic is also another contributor, seeing the paper by Villamil-Otero et al. (2017) above.

We agree and have changed the wording to be more explicit about this and included this reference here and for one other related discussion:

“...The Atlantic Ocean is a large area of relatively warm open water, serving as vital moisture and energy sources for cyclone development along the Atlantic storm-track into the Arctic [Timo et al., 2016; Villamil-Otero et al., 2018]. As shown in Villamil-Otero et al. [2018], Atlantic storms advect more heat and moisture into the Arctic than Pacific storms...”

“...Therefore, given the same cyclone occurrence, cyclones in the Lincoln Sea were more intense and accounted for a greater proportion of the total snowfall than cyclones in the Chukchi Sea in 1979-2016 (Figs. 2d-f and 3b). Based on previous works [Timo et al, 2016; Villamil-Otero et al., 2018], we attribute this to the greater capacity of Atlantic cyclones to advect more heat and moisture, as well as their larger size, greater strength, and longer duration, relative to their Pacific counterparts...”

10. Line 144-145: This could be also because the variability of snowfall caused by other factors is small.

This is an excellent point. We’ve included a discussion (shown in the following comment below) to help readers consider this aspect.

11. Line 145-148: There could be uncertainties in the statement in this sentence. This is because (1) seeing the comment on line 144-145 above; and (2) most of snow has fallen before the cyclone enters the interior Arctic (i.e., no water to support snowfall in the interior Arctic Ocean). *We agree and revised this content to be more explicit about some of the factors that likely influence the spatial patterns in inter-annual variability:*

“The spatial discrepancies in the variability between cyclone snowfall, snowfall contribution, count, and intensity point to a multitude of effects. For example, the variability in other snowfall mechanisms influence the cyclone snowfall contribution since the latter is a function of total snowfall. Furthermore, the available moisture and energy sources for cyclones are limited by sea ice, which inhibits heat and moisture uptake. Consequently, cyclone snowfall may be less in regions that have a thick, compacted sea-ice cover, which is especially relevant if most moisture is precipitated out before cyclones reach the interior of the sea-ice pack.”

12. Line 164: In the previous sentence, for the Arctic average, freeze-up date can explain ~20%. But the number changes to from ~61% in the Arctic Ocean. I would suggest the authors to clarify.

Yes, we can see how this is confusing. We revised the sentence to first state the average across all Arctic regions, and then go into a regional breakdown of the range in correlations. We found an error in our numbers and rectified it as well.

“Investigating the effects of freeze-up on snow accumulation further, we find that the average freeze-up date across all Arctic regions controls ~30% of the variability in annual snow accumulation (Fig. 5a; Supplementary Table 1). However, this relationship varies from ~0% in the Laptev Sea to 49% in the Barents Sea.”

13. Line 168-169: Snow accumulation is defined the difference between the first and last date of this period. Snow/ice melt unlikely occur in March. If there is an early melt in May, why is there is zero snow accumulation? This may need a clarification.

Thank you for identifying this issue. We looked into it and indeed found a problem that you suspected – the sea ice begins to disappear at the most southerly latitudes at the end of May, leading to negative snow accumulation values. To reconcile this, we took the maximum snow depth for the spring season (which tends to occur in early May) minus the snow depth from the first day of the spring season to yield spring snow accumulation. For the autumn and winter seasons, we still use difference in snow depth between the last and first date of the season to calculate snow accumulation. We have updated the statistics, tables, and figures accordingly.

14. Line 225-226: If snow depth is used as a measure to get a composite of cyclone frequency, intensity, and snowfall, I would like the author to provide a brief description how the composites were derived considering that snow depth reaches its criterion on different date at different grid points.

We expanded the description of the approach in the text for clarification. To summarize, we identify the 1979-2016 average date at which 25%, 50%, and 75% of the snow cover was established using the model snow depths. The date of maximum snow depth at a given location is used to back-calculate when 25%, 50%, and 75% of the maximum snow depth occurs. These “back-calculated” dates are subsequently used to composite the cyclone variables.

15. Line 242-244: The Lincoln Sea is covered by stable thick ice. Generally there is no clear dynamic and thermodynamic forcing for cyclone intensification there. It would be interesting and useful to discuss why cyclone intensity and snowfall can be larger in the Lincoln Sea than in the Chukchi Sea.

We added a discussion on this point in the text:

“Furthermore, the available moisture and energy sources for cyclones are limited by sea ice, which inhibits heat and moisture uptake. Consequently, cyclone snowfall may be less in regions that have a thick, compacted sea-ice cover, which is especially relevant if most moisture is precipitated out before cyclones reach the interior of the sea-ice pack.”

16. Line 255-257: This sentence may be revised to: The relatively higher correlation with the cyclone vorticity is because vorticity is a measure of strength of rotation and, in turn, convergence/uplift, which transport moisture and heat from open water surface into the atmosphere. The moisture can then precipitate as snow onto sea ice.

Thank you for the helpful suggestion. We revised it accordingly.

17. Line 283-284: Please refer to the paper by Villamil-Otero et al. (2017) mentioned above. *We removed this content to streamline and shorten the manuscript, but we do reference Villamil-Otero et al., [2017] earlier in the text when this topic is discussed. Thank you again for the informative reference.*

18. Line 319: The text below and the figure caption indicate there is no statistically significant trend. I suggest the authors to clarify what "exception" means here.

This section (Inter-decadal changes and variability) has been substantially revised following all reviewers' comments. We hope the content is much clearer and fully addresses your suggestion here.

19. Line 321-324: This is consistent with the recent finding of Beaufort high strengthening and autumn surface wind increase over the Beaufort and Chukchi seas:

Moore 2012: Decadal variability and a recent amplification of the summer Beaufort Sea High. *Geophys. Res. Lett.*, 39, L10807, doi:<https://doi.org/10.1029/2012GL051570>.

Wu, Q.; Zhang, J.; Zhang, X.; Tao, W. Interannual variability and long-term changes of atmospheric circulation over the Chukchi and Beaufort seas. *J. Clim.* 2014, 27, 4871–4889.

Stegall S T and Zhang J 2012 Wind field climatology, changes, and extremes in the Chukchi–Beaufort seas and Alaska north slope during 1979–2009 *J. Clim.* 25 8075–89.

Zhang, J., S. T. Stegall, and X. Zhang, 2018: Wind–sea surface temperature–sea ice relationship in the Chukchi–Beaufort Seas during autumn. *Environ. Res. Lett.*, 13, 034008, <https://doi.org/10.1088/1748-9326/aa9adb>.

Thank you very much for bringing these works to our attention. Although we've removed this specific content, these and other works are a big motivation for future work.

20. Line 326-328: Decrease in cyclone-induced snowfall can also cause a decrease in its contribution even if the snowfall by other systems does not change. The decrease in cyclone-induced snowfall is consistent with the decrease in cyclone frequency and intensity, as well as

strengthening of the Beaufort high as mentioned above. Considering this, I suggest to revise the next sentence, because there is no evidence of other system-caused snowfall increase in current study.

We agree and have revised all of the content in this section. We also added discussions throughout the manuscript on the factors that you raised in your review that may influence the cyclone snowfall contribution (which are listed in our previous responses to your comments). Thank you for the constructive feedback.

21. Line 338: I would suggest revising "has caused" to "may have greatly caused", because of effect of sea ice convergence to this area.

We have since removed this content to place more focus on statistically significant findings.

22. Line 752: Figure 9 should be Figure 10.

Thank you for the correction. We've revised the figure numbers based on the updates made to figures in the main text and in supplementary information.

Reviewer #3 (Remarks to the Author):

Recommendation: major revision

Review by Paul Kushner, University of Toronto

General comments

1. In this study, the authors investigate controls exerted by cyclones on snow cover on Arctic sea ice. They simultaneously characterize Arctic cyclone tracks and activity, their related snowfall characteristics, and snow depth, examining climatology, climate variability, and trends of these quantities. Data sources include reanalysis, satellite derived snowfall, ice mass balance/motion/concentration data. The authors quantify Arctic-cyclone contributions to total snowfall and infer contributions to snow depth, across regions and seasons of the Arctic. Analysis is also devoted to understanding sensitivities to different characteristics of cyclones, including frequency and depth. Cyclone activity explains about half and sea ice freeze up timing explains about a fifth of the variance in snow depth, and strong regional and seasonal variations across the Arctic are documented. The findings are characterized as heterogeneous and complex, and are taken as a starting point for better understanding of the mechanisms of snow on sea ice. *We couldn't have summarized it better ourselves. Thank you.*

2. This ambitious and original study deals with a cutting edge topic in Arctic and global climate science. As articulated in this study and in other publications (including several from led by this author team), the characteristics of snow on sea ice controls important sea ice characteristics and provides a major observational constraint for retrieval of sea ice thickness from available satellite observations. The contributions of extratropical cyclones to Arctic snow on sea ice has not previously been studied, to my knowledge, in such a comprehensive manner. The study makes clear the challenge of characterizing Arctic cyclones and then quantitatively elucidating their contribution to snowfall on sea ice. It broadly provides a very strong basis for future work, and advances understanding of this tricky area considerably. Key figures provide an excellent

visualization of the patterns and complexities: Fig. 2 showing the Arctic cyclone and snowfall characteristics, Fig. 3 showing the snow depth reconstruction, Figs. 6-7 giving an overview of the seasonal evolution of the snow-on-sea-ice impact, and Fig. 8 showing revealing case studies. The background for the paper in terms of the previous literature is excellent, the write up generally clear, and evidence provided for key conclusions is generally persuasive.

Characterizing Arctic cyclone impact on snow on sea ice is an interdisciplinary topic requiring input from both atmospheric and cryospheric science as well as a mastery of multiple observational datasets. Given this and the importance of the topic, the material written up in this study would be of interest and value to the atmosphere, cryosphere, and climate science community.

Thank you for the kind summary.

3. That being said, the current manuscript has weaknesses that lead to my recommendation of "major revision": the study is a bit too long (does it satisfy Nature Communications word limits?), the write up is repetitive and loose, there are mistakes in the document preparation, and some conclusions are only weakly supported by the presented evidence. Fig. 3 is potentially excellent but currently confusing, Fig. 4 is hardly referred to at all, and there is too much space devoted to the statistically insignificant trends found in Figs. 9 and 10 (which is captioned as Fig. 9). The material in supplementary information is not well documented and the use of this material is unclear. Sleuthing skills are required to make it through the current manuscript. With the exception of the trend analysis, these concerns can be addressed without changing the important conclusions, and the recommendations here are intended to increase the impact of the work.

Thank you so very much for the helpful feedback. We agree with all of your points and are deeply appreciative of your time and effort in providing valuable suggestions. Following your suggestions here:

- *We shortened the manuscript's length. It was indeed too long. We cut 2,000 words (including the new statistics section) to meet the Nature Communications word limit.*
- *We removed redundancy throughout the text, consolidated discussions where appropriate, and streamlined the flow of the results and discussion topics.*
- *We reconciled mistakes in the document preparation. We apologize for the sloppiness. We originally intended to submit to a different journal, but were encouraged to first submit to Nature Communications beforehand.*
- *We agree Figure 3 was confusing before. We've made changes to make it easier to interpret, such as lighter shades consistently representing higher values and different colors to better distinguish variables.*
- *We removed Figure 4 and used that space to create a figure (now Figure 5) of correlations from the univariate and multivariate analysis to bring more transparency to the results and strengthen the conclusions.*
- *The trend section has been revamped and shortened to emphasize the key, statistically-significant findings.*
- *We ironed out supplementary information to include only what is most relevant to the analysis and moved some material from the supplementary information into the main text to make it easier for readers to interpret the results.*

4. The statistical analysis is not well explained and leaves unopened questions.

The authors need to be more explicit about the physical-statistic models they have in mind with all the possible control variables. The statistical analysis starts sneaking in at 1.163, where it is claimed that 20% of the variance comes from freeze-up date. Further digging shows that this is found in the supplementary information, Table S2.

Similarly, e.g. at 1.252 statements related to ranges variance explained come from the tables in the supplementary information - these should be referenced.

There is insufficient explicit recognition in the text that several of the processes/mechanisms studied could be cross correlated, including interannual variability in cyclone frequency and intensity, cyclone activity and sea-ice freeze up dates, etc., which could affect interpretation. This is hinted at starting 1.267 but not really stated explicitly.

There is no explanation of how r^2 of as small as 0.03 ($r \sim 0.17$) could be statistical significant (e.g. Table S3). I assume the authors are assuming zero serial correlation for a 38 year time series, but is this correct, i.e. are all years statistically independent? There is no explanation of the trend analysis carried out and whether it is assuming linear trends in all fields, which might be naive.

We agree that the details on the statistical analysis were sorely lacking. The statistical analysis has been bolstered by several elements in the revision:

- *Most notably, the addition of a new section (Statistical Approach, see below) goes into rich detail of the statistical models used, the assumptions made, and justification for assumptions. The assumptions are backed up by supporting figures in the supplementary information, which are now directly referenced in the main text and methods section. We also included a discussion in supplementary materials.*
- *We created a new figure (Figure 5) from the tables in supplementary information to increase transparency of the statistical results, as well as make it considerably easier to interpret the results.*
- *We now directly reference Figure 5 and supplementary figures/tables in the main text.*
- *We tested linear, cubic, and exponential models. Linear regressions yielded the lowest root-mean-squared (RMS) errors for the majority of cases, demonstrating the strongest fit among variables. The exceptions were total snowfall and total precipitation, which cubic and exponential models yielded the lowest RMS errors, respectively. For consistency across variables and regions, we deferred to using a linear regression model.*
- *We include an in-depth discussion on the aspect of cross-correlations between variables in the main text and in supplementary materials, which is supported by new figures.*
- *Specific to the freeze-up analysis, we carried out a Durbin-Watson test to determine the possibility of autocorrelation among variables in the 37-year time-series. We found no autocorrelation with the exception of cyclone snowfall in the E. Siberian Sea and total precipitation in the Arctic Ocean (autocorrelated); total snowfall in the Arctic Ocean and Laptev Sea and vorticity in the Arctic Ocean (inconclusive). These results establish that, with few exceptions, the assumption of a zero serial correlation is justifiable for variables in the 37-year time-series. For snow depth, this assumption may not hold in reality since some*

snow survived the summer melt season in earlier decades [Radionov et al., 1997]. However, for our modeled snow depth, we use zeros for the initial snow conditions for each September, which removes any possibility of there being a carryover of snow from the preceding year and creating an artificial trend in annual snow depths, and subsequently makes a zero serial correlation suitable.

“Statistical Approach

To evaluate the relationship between cyclones and snow accumulation on sea ice, we used univariate linear regressions between the monthly snow accumulation (dependent) and monthly cyclone characteristics (independent) over sea ice. To validate the choice of a linear regression model, we examined linear, cubic, and exponential regressions for all variables across all regions. Linear regressions yielded the lowest root-mean-squared (RMS) errors for the majority of cases (see Supplementary Information), demonstrating the strongest fit among variables. For consistency across variables and regions, we deferred to using a linear regression model. The analysis includes all variables in September-May for 1979-2016, has a sample size of 336, and here and for all other statistical analyses conducted, 0.05 is used for statistical significance. A monthly temporal resolution was chosen to avoid the high-frequency fluctuations in daily snow depth due to ice motion. The cyclone characteristics are calculated as the monthly regional mean of the sum of: (a) count; (b) vorticity; (c) size; and (d) snowfall over sea ice. The monthly sum is taken from the last day of the preceding month to the penultimate day of the relevant month. This one-day offset accounts for the lag of snow accumulation on sea ice following cyclone events.

Using monthly variables over sea ice, we further evaluated the combined effect of cyclone characteristics on snow in a multivariate framework using a sample size of 336. The multivariate models use monthly snow accumulation (dependent) and (a) cyclone count and vorticity; and (b) count, vorticity, and size (independent). The models use the regional mean of the temporal sum of cyclone count, but the regional mean of the temporal mean in cyclone vorticity and size to avoid double counting cyclones and erroneously estimating their effect on snow accumulation. Scatterplots in Supplementary Figure 5c,g-h demonstrate that the cyclone variables considered in the multiple regression framework largely do not exhibit collinearity. This suggests that more cyclone events do not equate to stronger or larger cyclones and establishes that these variables can be treated as independents. Cyclone snowfall was not included in the multivariate analysis to avoid multicollinearity (Supplementary Fig. 5a,e)

Given the strong positive correlation between cyclone snowfall and snow accumulation (see Fig. 5b and Supplementary Table 2), we analyzed the relationship between cyclone count and vorticity and cyclone-associated snowfall. Regional averages of the monthly sums in these variables were used, resulting in a sample size of 456. Supplementary Figure 5a,e demonstrates the positive correlations between cyclone snowfall and count and vorticity and a strong fit of the linear regression model.

Prior to the trend analysis, we tested for autocorrelation among variables using the Durbin-Watson Test. The test reveals that, despite a few exceptions, cyclone counts, cyclone snowfall, vorticity, total snowfall, and total precipitation are not autocorrelated. The exceptions are: cyclone snowfall in the E. Siberian Sea and total precipitation in the Arctic Ocean (autocorrelated); total snowfall in the Arctic Ocean and Laptev Sea and vorticity in the Arctic Ocean (inconclusive). Furthermore, we use zero initial values in the snow depth reconstruction

each year, which removes any carryover of snow from the preceding year. Therefore, with a few exceptions, the assumption of a zero serial correlation is justifiable for variables in the 37-year time-series. We examined linear, cubic, and exponential regressions for all variables across regions and chose linear regressions due to the strongest fits for exploring whether discernable trends in variables exist for 1979-2016. We note that, for several variables, the trends were inconclusive. We found no significant or coherent relationships between freeze-up and cyclone characteristics. Collinearity between these variables is unlikely to affect the interpretation of the linear regressions and interdecadal trends.”

In supplementary materials:

“Cyclone variables such as count and snowfall can co-vary (Supplementary Fig. 5a,e). However, the correlations are regionally dependent and it is important to understand the influence of each cyclone variable on snow accumulation in a univariate framework. Our analysis of the data characteristics suggests that cyclone count, vorticity, and size largely do not exhibit collinearity (Supplementary Fig. 5c,g,h). Therefore, in the climatology, more cyclone events do not equate to stronger or larger cyclones and these variables can be considered in a multivariate framework. Prevailing atmospheric conditions and oscillations may influence interannual variability in cyclone characteristics. However, the univariate and multivariate regression analysis does not consider the correlation in cyclone variables over time, but explores the relationships between individual monthly measures of cyclone characteristics and the snow accumulation. While some resulting correlations are small, they are still statistically significant given the large sample size (336 or 456). Nevertheless, we note that some individual cyclone characteristics do not have strong relationships with snow accumulation given the small correlations. For the trend analysis, we explored linear, cubic, and exponential regressions for all variables across regions. Linear regressions exhibited the strongest fits with the data (i.e. lowest root-mean-squared errors) with only two exceptions. The exceptions were total snowfall and total precipitation, which cubic and exponential models yielded the lowest RMS errors, respectively. Given that the scope of the analysis does not focus on trends in total snowfall or total precipitation, we deferred to using linear regressions for all variables in the trend analysis for consistency in approach.”

5. ‘Interdecadal changes and variability’: A lot of space is devoted (1.315-366) to a set of trends maps in Fig. 9 that are not statistically significant - this section is not persuasive. It is worth noting but perhaps not surprising that these noisy patterns end up having consistent signals across variables, but that doesn’t make them more significant as temporal signals. A better starting point is Figure 10 (mis-captioned as Fig 9 at 1.152) which shows the very interesting pattern of increasing and decreasing snow depth. At least here there are large regions with >90% significance. But here again, trends are emphasized that are not significant (1.340). Perhaps instead of the seasonal breakdown shown in Fig. 9 the authors could ask if there are trends in accumulated snowfall over the full snow season to help explain Fig. 9. Would spatial aggregation over the different regions help, instead of maps of local trends? E.g. to really cherry pick the authors could calculate area-mean snowfall/cyclone trends at those locations where the snow depth trends are significant.

This feedback was extremely helpful – thank you. We revised this section entirely following yours and other reviewers’ suggestions:

- We cut the content down considerably.

- *We placed snow depth as the guiding focus of this section. We first present content on the positive trends in snow depth, followed by results on the negative trends.*
- *We remove the description of seasonality. We agree that there wasn't enough substance there due to the statistically insignificant findings. We used annual values instead.*
- *We used regional aggregations rather than maps of local trends for the results.*
- *We put the resulting values in a referenced table in the main text.*
- *We opted not to cherry-pick since the areal fraction where snow depth trends were significant was quite small.*
- *We used 95% statistical significance when redoing the trend analysis for consistency across all statistical analyses.*

6. For a contribution to the high impact literature, the material lacks punch and could be tightened. E.g. lines 24-29 in the abstract mirror the first couple of paragraphs in the introduction, some sentences seem to be repeated almost word for word. Also, I wonder why the authors provide the summary in the abstract and again in the introduction (lines 76-94), instead of presenting evidence and using a concluding section to highlight key points before discussing them. Another example: the sea ice freeze up point is introduced twice, once starting at line 157 and again starting at line 271.

We agree and have substantially streamlined the content.

7. Has this work been prepared for some other resource or reference? There are erroneous Section references to things like "Section 3.2" and "Section 4.4.2" at lines 117, 286, 362, 478, 484.

Thank you for identifying these. The mistakes have been corrected.

8. The peak in cyclonic activity in autumn on the Pacific side (starting at 1.173) is hard to see in Fig. 2. Since this is listed as a key conclusion (1.79) more evidence should be provided. Could it be supplemented with the full seasonal cycle of cyclonic activity as a time series, including summer, for both Pacific and Atlantic sectors, eg as a supplementary figure? This would provide context related to the documented summer cyclone maximum from Serreze and Barrett 2008.

Yes! That's a figure that has been added, and we find it to be much more helpful for referencing and putting this work into context with others. Thanks for the excellent recommendation.

9. Points about the figures:

Figure 3: It is not clear why the shading scale is reversed in the two last columns of Fig. 3. In all panels in Fig. 3, is the region south west of the Bering Strait being plotted properly? It is not clear how a snow accumulation of zero at the edges of e.g. Figure 3F and can be accompanied by maxima of Std Dev of accumulation in e.g. Fig. 3I. Can accumulation have a large standard deviation if it has a zero mean? The related discussion at lines 168-171 is confusing. Perhaps mask with a distinctive masking color regions in 3D-F that are near zero in 3G-I.

There is no good reason why the shading scale is reversed. We have corrected this. Regarding the zero accumulation, we looked into it and indeed found a problem – the sea ice begins to disappear at the most southerly latitudes at the end of May, leading to negative snow accumulation values. To reconcile this, we took the maximum snow depth for the spring season (which tends to occur in early May) minus the snow depth from the first day of the spring season

to yield spring snow accumulation. For the autumn and winter seasons, we still use difference in snow depth between the last and first date of the season to calculate snow accumulation. We have updated the statistics, tables, and figures accordingly.

Fig 4 is mentioned in passing once and not brought up again. This could be replaced by a figure that shows the seasonal cycle in time series form as suggested above.
 Thank you for the excellent suggestion. We made these changes.

Fig 9 is stated as having only not statistically significant trends, but at lines 319 and 333 there are significant trends discussed. Not sure a localized trend would qualify as ‘field significant’. In any case, are there significant trends anywhere in Fig. 9?
 We should have used hatching for Figure 9 to make significance more apparent. We have since removed this figure and revised the content in this section to focus on regional, annual trends. There are some statistically significant trends, but not many with regard to cyclone variables (see new Table 1 that’s now in the main text):

	Snow Depth (cm)	Cyclone Count (#)	Cyclone Vorticity (hPa)	Cyclone Rainfall (mm)	Cyclone Snowfall (mm)	Cyclone Precip. (mm)	Total Rainfall (mm)	Total Snowfall (mm)	Total Precip. (mm)	Early Freeze-up (days)	Late Freeze-up (days)
Arctic O.	-0.2	4.5	0.079	0.001*	0.001	0.002*	0.001*	0.001*	0.002*^A	18*	16*
Barents	-3.8*	1.9	-0.009	0.006*	-0.005	0.001	0.006*	-0.004*	0.002	14*	2
Beaufort	-0.8*	2.1	0.021	0.003	0.000	0.003	0.004	0.000	0.004	14*	15*
Chukchi	-1.7*	0.8	0.007	0.004	0.001	0.005	0.006*	0.003	0.010*	18*	16*
E. Greenland	2.4*	0.6	-0.023	0.004	-0.004	0.000	0.005*	-0.004*	0.001	0	0
E. Siberian	-1.5*	4.6	0.022	0.002	-0.001* ^A	0.001	0.003*	-0.000	0.003	8*	7
Kara	-1.9*	-6.5	-0.098	0.003	0.001	0.004	0.004	0.004	0.008*	4	3
Laptev	-0.7*	-3.1	-0.033	0.004	0.004	0.007	0.004	0.006	0.009	3*	2
Lincoln	0.5	6.5	0.119	0.002*	0.005	0.006*	0.002*	0.005	0.007*	1	0

Table 1 | The 1979-2016 regional trends in snow depth, cyclone variables, precipitation, and sea-ice freeze-up per decade. Bold text and asterisks indicate at least 95% statistical significance. Rainfall, snowfall, and precipitation amounts were normalized by the corresponding regional areas to allow inter-comparison across regions. Note, cyclone and precipitation variables were not normalized by the sea-ice area due to the negative trends in sea-ice extent. The superscript “A” indicates variables subject to autocorrelation.

Figures with maps use several different conventions for describing panels and columns. E.g. ‘Column 1’ versus ‘first column’ versus no reference to columns. The authors should choose one convention.

Thank you; we’ve chosen one convention and have stuck with it.

10. At a couple of points, e.g. l.36, the contrast in regionality of sea-ice freeze up controls and cyclone activity is emphasized. Cyclone activity r^2 range from 10% (Barents) to 83% (Lincoln) according to Table S3 last column, whereas sea-ice freeze up range from 11% (Barents) to 61% (Arctic Ocean), which is less of a range than cyclone activity, according to Table S2. So looking at these tables it seems there’s as much heterogeneity from cyclone activity as from sea-ice freeze up.

Yes, we agree and have adjusted the wording of the conclusions accordingly to emphasize that there is much heterogeneity from cyclone activity as from sea-ice freeze-up. We also found an error in the freeze-up correlations and rectified it (in short, it still shows strong regionality).

11. The idea of snow carried on exported sea ice appears at 1.288 as a possible mechanism, where it is not listed previously to this. Why does the ice have to come from ‘more northerly latitudes’? *We clarified this point in the text to reference to snow conditions in the Beaufort Sea. Multiyear sea ice is regularly advected from north of the Canadian Archipelago and the Lincoln Sea, bringing with it deeper snow. The “tongue” of multiyear ice is often apparent in Operation IceBridge snow depth retrievals in the Beaufort Sea.*

12. The story could be made more interesting by starting with something like Figure 8 which shows contrasting behaviour of snow build up in two regions, also highlighting the excellent use of IMB’s here, and then generalizing to the other figures. This is just a suggestion, it’s up to the authors whether to pursue it.

We mulled over this suggestion for an embarrassing amount of time. We very much like it, but felt that it was too much to restructure the manuscript to such a large extent. That said, it really is a fantastic idea and we appreciate the suggestion. We’ll incorporate it into future presentations.

Minor comments

1. 1.24: complex*, *

Fixed – thank you.

2. 1.36: ‘cyclone activity explained’ is vague

Reconciled. We use more descriptive wording throughout the text.

3. 1.37: overlaps with lines 31-34

Redundancy removed; thank you.

4. ‘Other snowfall mechanisms’ phrase and related text is repeated too many times in the ms, e.g. 1.85, 1.116, 1.362, 1.381, etc.

We agree. We removed this content and made the description more detailed (once).

5. 1.46: introduction repeats too much of intro material

We agree and adjusted the text accordingly.

6. 1.63: previous *work has not*

Thank you; fixed.

7. 1.65: use of word ‘models’ suggests that authors will look at models here.

We added “snow” and “reanalysis” to be more description throughout the text.

8. 1.78: delete ‘bimodal’ - unsuitable term. Suggest: ‘... regional differences and, in particular, cyclone activity peaks in winter in the Atlantic sector and peaks in autumn in the Pacific sector.’ (Is this conclusion really original, given the previous literature?)

We deleted this material out of redundancy, but we agree, it's not the key, original finding from this work.

9. 1.135 forward: the variation in quantities with the ice margin variability is not fully evident in the figure.

We've revised this content entirely. We agree that the strong variability wasn't fully evident before.

10. 1.161: 'mute' - do the authors mean 'overwhelm' or 'attenuate'?

Yes, thank you.

11. 1.195: delete 'in this region'

Deleted. Thanks.

12. 1.232: cyclone 'density' used here and elsewhere but never defined

Following yours and other reviewers' suggestions, we went with one convention to describe cyclone events, using "count" or "event count" rather than density. We also included a clear description of cyclone count.

13. 1.249: seasonal 'cycle'

Yes, thank you.

14. 1.310: should be Figure 7 and *8*?

Yes, we've revised all figure numbers as well to correspond with the recent changes. They should be consistent throughout the manuscript now.

15. 1.326: This discusses a marginal feature - should it be kept? In any case the wording is awkward. Suggest 'there is a decrease that is much smaller in magnitude or a slight increase over the ...'

No, we feel it shouldn't be kept. Thank you for your thoughts on this.

16. 1.384: explain up to ~83% . . . -> explain, regionally, over 80%

Revised.

17. 1.401: obscure connection to Fig. S1

We kept in this figure since it's included in the discussion section, but removed its reference here.

18. 1.502: No initial conditions -> Initial values of zero

Revised. Good point.

19. 1.504: black->blue

Corrected. Thank you.

20. 1.696: 'the reconstructed snow depth' - not clear what this is, refer to methods.

We've tried to make this more explicit in the revision. The snow depths are referred to as snow

depth reconstructions, or reconstructed snow depths. We opted to refer to it this way for consistency with the literature (Kwok et al., 2017; Blanchard-Wrigglesworth et al., 2018).

21. 1.748: does this sentence require revision given comment above?

This figure was deleted and converted into a table showing regional values. The related sentences were revised.

REVIEWERS' COMMENTS:

Reviewer #1 (Remarks to the Author):

First, I want to commend the authors on undergoing additional analysis and significant modification to the text in order to address reviewer comments. From my perspective, the authors sometimes did more than would be expected to either improve or justify the work. The diligence is great! I have a few comments to make at this stage, one of which I think is good for science but pushes beyond the scope of the current paper.

Some Line-by-Line Comments

Lines 117 – 123: Reading through this now, the meaning of point (c) and how it differs from (a) takes some mental work. I'm pretty sure the authors mean in (c) that later freeze-up causes a shift in the seasonal snowfall cycle. If that's the case, maybe using the word "cause" would help clarify. If not, then I suppose a different clarification would also be helpful.

Line 142: "Cyclone events" is a discrete count, so should that be "fewer" instead of "less"

I think that Lines 32-33 in the Abstract ("While cyclones are stronger in the Atlantic, Pacific snow accumulation is more sensitive to cyclone strength. ") is an important finding because we're often concerned with the variability rather than the mean in understanding how to predict a system. Supplemental Table 2 is a very good support for this statement, but I can't find this connection made in the results sections in which I'd expect to find it (somewhere in Lines 125-194).

Line 264: Back on Line 72, the phrasing is "~80%", but here it's "more than 80%"). Looking at Figure 2, I think the "about 80%" is the more accurate statement, because that's the maximum for the color bar, and taking the Atlantic Sector overall, it's not all saturated at 80% in the Dec-Feb plot and hardly if ever in the Sep-Nov and Mar-May plots. So really, "up to 80%" might be the most accurate for both places. Either way, it's well over 2/3, so it's high.

Table 1: The "7" for East Siberian Sea Late Freeze-Up is bold but has no asterisk.

On the cyclone-associated precipitation measures...

The point of bringing up what I did in the initial review was meant to highlight a choice in the detection method that is sometimes overlooked but which can impact calculations like the percentage of snowfall being associated with storms. I did say "it might be worth testing" in the first review, but I also said, "I think the authors could also improve handling of bias simply by enriching the discussion section." That last statement is what I want to emphasize right now. I can see the authors have done this adequately in the discussion section, so that should be fine in my view.

However, now that the authors have implemented some additional work, I feel compelled to have a conversation about it. The figure provided showing maps of the difference between the method implemented by the authors and the Finnis/Stroeve method doesn't quite make sense since the Finnis/Stroeve example seems to have a circle of precipitation... The main benefit of that method is that it explicitly seeks to identify non-symmetrical spatial patterns by recording all of the precipitation for any coherent precipitation region if any of that region intersects the cyclone area. In other words, it's meant to inclusively take the union of the cyclone area and any intersecting precipitation area, not exclusively the intersection of the areas. Looking at the figure, I wonder if that might be why the Finnis/Stroeve method precipitation ended up as it is, and if so, it's not fully implemented. Anyway, I agree with the authors that this is a source of variance between studies that merits further investigation.

Reviewer #2 (Remarks to the Author):

Review of the manuscript entitled "The role of cyclone activity in snow accumulation on Arctic sea ice" by Webster et al.

The authors have adequately addressed my comments and suggestions and very carefully revised the manuscript. The revised manuscript therefore presents solid results with much reduced uncertainties. I appreciate the authors' effort and would recommend publication of this excellent manuscript in its current form.

Xiangdong Zhang

Reviewer #3 (Remarks to the Author):

The authors have thoroughly addressed my concerns from the earlier manuscript and in particular have bolstered the description of the analysis which will make the results more straightforward to understand. I recommend publication.

A couple of comments:

- Please review the caption of Figure 1 - the colors mentioned, dots, etc. do not quite align with the caption.
- Cyclone vorticity is only defined through the citation (line 351) and units are not provided e.g. in Fig. 3b. More detail here would be useful since vorticity of a cyclone is ambiguous compared to SLP of a cyclone.
- Line 69: unclear what is meant by 'a similar spatial distribution' - are we saying a homogeneous spatial distribution?
- The discussion in lines 91-103 is a bit loose. Generally, it is not clear to me what we should expect of relationships between these kinds of cyclone and precipitation statistics in any sector, not just the Arctic. It is sufficient to point out that there is no obvious relationship without speculating on effects such as moisture availability (e.g. lines 100-103) - but perhaps the authors have support for this claim from other papers that weren't cited here.

REVIEWERS' COMMENTS:

Reviewer #1 (Remarks to the Author):

First, I want to commend the authors on undergoing additional analysis and significant modification to the text in order to address reviewer comments. From my perspective, the authors sometimes did more than would be expected to either improve or justify the work. The diligence is great! I have a few comments to make at this stage, one of which I think is good for science but pushes beyond the scope of the current paper.

Thank you very much for your time and energy in helping us improve the analysis. We greatly appreciate it!

Some Line-by-Line Comments

Lines 117 – 123: Reading through this now, the meaning of point (c) and how it differs from (a) takes some mental work. I'm pretty sure the authors mean in (c) that later freeze-up causes a shift in the seasonal snowfall cycle. If that's the case, maybe using the word "cause" would help clarify. If not, then I suppose a different clarification would also be helpful.

Thank you; we incorporated this suggestion.

Line 142: "Cyclone events" is a discrete count, so should that be "fewer" instead of "less"

Good catch – thank you!

I think that Lines 32-33 in the Abstract ("While cyclones are stronger in the Atlantic, Pacific snow accumulation is more sensitive to cyclone strength.") is an important finding because we're often concerned with the variability rather than the mean in

understanding how to predict a system. Supplemental Table 2 is a very good support for this statement, but I can't find this connection made in the results sections in which I'd expect to find it (somewhere in Lines 125-194).

Thank you for the suggestion. We emphasized these results in lines 185-201 and again in the discussion (lines 302-305).

Line 264: Back on Line 72, the phrasing is “~80%”, but here it's “more than 80%). Looking at Figure 2, I think the “about 80%” is the more accurate statement, because that's the maximum for the color bar, and taking the Atlantic Sector overall, it's not all saturated at 80% in the Dec-Feb plot and hardly if ever in the Sep-Nov and Mar-May plots. So really, “up to 80%” might be the most accurate for both places. Either way, it's well over 2/3, so it's high.

Right. We changed this before to more than 80% based on another Reviewer's suggestion. We've gone with the exact percentage this time to address both your and the other reviewer's suggestions.

Table 1: The “7” for East Siberian Sea Late Freeze-Up is bold but has no asterisk.

Good catch! Thank you.

On the cyclone-associated precipitation measures...

The point of bringing up what I did in the initial review was meant to highlight a choice in the detection method that is sometimes overlooked but which can impact calculations like the percentage of snowfall being associated with storms. I did say “it might be worth testing” in the first review, but I also said, “I think the authors could also improve handling of bias simply by enriching the discussion section.” That last statement is what I want to emphasize right now. I can see the authors have done this adequately in the discussion section, so that should be fine in my view.

Thank you for the suggestions. We greatly appreciate the resources for other approaches, as we can learn much from them and apply aspects of them in ongoing and future analyses.

However, now that the authors have implemented some additional work, I feel compelled to have a conversation about it. The figure provided showing maps of the difference between the method implemented by the authors and the Finnis/Stroeve method doesn't quite make sense since the Finnis/Stroeve example seems to have a circle of precipitation... The main benefit of that method is that it explicitly seeks to identify non-symmetrical spatial patterns by recording all of the precipitation for any coherent precipitation region if any of that region intersects the cyclone area. In other words, it's meant to inclusively take the union of the cyclone area and any intersecting precipitation area, not exclusively the intersection of the areas. Looking at the figure, I wonder if that might be why the Finnis/Stroeve method precipitation ended up as it is, and if so, it's not fully implemented. Anyway, I agree with the authors that this is a

source of variance between studies that merits further investigation.

Thank you for following up on this discussion. We greatly appreciate the clarification on the Finnis approach. We suspect the circle of precipitation came from a boundary that was applied in Stroeve et al. (2011), which we mistakenly used as an intersection rather than a union. They state that:

“We identify cyclone-associated precipitation using the same approach as Finnis et al... ..and define cyclone-associated precipitation at all grid points that falls within a 250 km radius of a cyclone centre (approximately one-fourth Rossby radius), the position of which is determined from the detection and tracking algorithm just described, and for which the largescale precipitation rate exceeds 1.5 mm d^{-1} , plus the convective precipitation within the same radius.”

We appreciate the benefit of taking the union of cyclone and precipitation areas as you pointed out. We plan to incorporate this approach in ongoing and future work, but unfortunately, would be too much to incorporate into this paper. Again, thank you for following up on this discussion. We hope that there will be an opportunity to discuss this topic further with you in the future.

Reviewer #2 (Remarks to the Author):

Review of the manuscript entitled “The role of cyclone activity in snow accumulation on Arctic sea ice” by Webster et al.

The authors have adequately addressed my comments and suggestions and very carefully revised the manuscript. The revised manuscript therefore presents solid results with much reduced uncertainties. I appreciate the authors' effort and would recommend publication of this excellent manuscript in its current form.

Xiangdong Zhang

Thank you very much for your time in carefully reviewing our work and providing constructive suggestions for improving the analysis. We very much appreciate it!

Reviewer #3 (Remarks to the Author):

The authors have thoroughly addressed my concerns from the earlier manuscript and in particular have bolstered the description of the analysis which will make the results more straightforward to understand. I recommend publication.

Thank you so much for carefully identifying weaknesses in this work and, even more, providing ways to strengthen them. We appreciate that took considerable time, and are

grateful for your help in improving the analysis. Thank you!

A couple of comments:

- Please review the caption of Figure 1 - the colors mentioned, dots, etc. do not quite align with the caption.

Good catch. Thank you.

- Cyclone vorticity is only defined through the citation (line 351) and units are not provided e.g. in Fig. 3b. More detail here would be useful since vorticity of a cyclone is ambiguous compared to SLP of a cyclone.

Right. We've provided clarification in the methods and added units to the figure labels in Figures 3b, 5, and Supplementary Figure 5.

- Line 69: unclear what is meant by 'a similar spatial distribution' - are we saying a homogeneous spatial distribution?

Good point; we expanded the description to:

"The basin-scale pattern in cyclone-associated snowfall (Fig. 2g-i), together with regional differences in sea ice conditions, creates a similar spatial distribution in snow depth on sea ice, with more snowfall and deeper snow towards the Atlantic and less snowfall and thinner snow towards the Pacific (Fig. 4a-c)."

- The discussion in lines 91-103 is a bit loose. Generally, it is not clear to me what we should expect of relationships between these kinds of cyclone and precipitation statistics in any sector, not just the Arctic. It is sufficient to point out that there is no obvious relationship without speculating on effects such as moisture availability (e.g. lines 100-103) - but perhaps the authors have support for this claim from other papers that weren't cited here.

We agree, and consolidated the content here especially since we discuss moisture availability (with references) in the preceding paragraph. Thank you for the suggestion.